# GluA3 subunits are required for appropriate assembly of AMPAR GluA2 and GluA4 subunits on cochlear afferent synapses and for presynaptic ribbon modiolar–pillar morphology

**Mark A Rutherford[1], Atri Bhattacharyya[1], Maolei Xiao[1], Hou-Ming Cai[2], Indra Pal[2], Maria Eulalia Rubio[2,3]\***

[1]Department of Otolaryngology, Washington University School of Medicine, St Louis, United States; [2]Department of Neurobiology, University of Pittsburgh School of Medicine, Pittsburgh, United States; [3]Department of Otolaryngology, University of Pittsburgh School of Medicine, Pittsburgh, United States

*For correspondence: mer@pitt.edu

**Competing interest:** The authors declare that no competing interests exist.

**Abstract** Cochlear sound encoding depends on α-amino-3-hydroxy-5-methyl-4-isoxazole propionic acid receptors (AMPARs), but reliance on specific pore-forming subunits is unknown. With 5-week-old male C57BL/6J *Gria3*-knockout mice (i.e., subunit GluA3$^{KO}$) we determined cochlear function, synapse ultrastructure, and AMPAR molecular anatomy at ribbon synapses between inner hair cells (IHCs) and spiral ganglion neurons. GluA3$^{KO}$ and wild-type (GluA3$^{WT}$) mice reared in ambient sound pressure level (SPL) of 55–75 dB had similar auditory brainstem response (ABR) thresholds, wave-1 amplitudes, and latencies. Postsynaptic densities (PSDs), presynaptic ribbons, and synaptic vesicle sizes were all larger on the modiolar side of the IHCs from GluA3$^{WT}$, but not GluA3$^{KO}$, demonstrating GluA3 is required for modiolar–pillar synapse differentiation. Presynaptic ribbons juxtaposed with postsynaptic GluA2/4 subunits were similar in quantity, however, lone ribbons were more frequent in GluA3$^{KO}$ and GluA2-lacking synapses were observed only in GluA3$^{KO}$. GluA2 and GluA4 immunofluorescence volumes were smaller on the pillar side than the modiolar side in GluA3$^{KO}$, despite increased pillar-side PSD size. Overall, the fluorescent puncta volumes of GluA2 and GluA4 were smaller in GluA3$^{KO}$ than GluA3$^{WT}$. However, GluA3$^{KO}$ contained less GluA2 and greater GluA4 immunofluorescence intensity relative to GluA3$^{WT}$ (threefold greater mean GluA4:GluA2 ratio). Thus, GluA3 is essential in development, as germline disruption of *Gria3* caused anatomical synapse pathology before cochlear output became symptomatic by ABR. We propose the hearing loss in older male GluA3$^{KO}$ mice results from progressive synaptopathy evident in 5-week-old mice as decreased abundance of GluA2 subunits and an increase in GluA2-lacking, GluA4-monomeric Ca$^{2+}$-permeable AMPARs.

## Editor's evaluation

Hearing is mediated by hair cells in the cochlea, which synapse onto the primary dendrites of the auditory nerve. This study shows how deletion of a postsynaptic glutamate receptor subtype strongly influences inner hair cell-spiral ganglion cell synapse formation. This work shows that pre- and post-synaptic changes intertwine dynamically, providing insights into how pathological outcomes arise from synaptic perturbations.

## Introduction

In the cochlea and ascending central auditory system, hearing relies on fast excitatory synaptic transmission via unique α-amino-3-hydroxy-5-methyl-4-isoxazole propionic acid receptors (AMPARs) (*Raman et al., 1994*; *Ruel et al., 1999*; *Gardner et al., 1999*; *Glowatzki and Fuchs, 2002*). AMPARs are tetrameric ionotropic receptor channels comprised of GluA1–4 pore-forming subunits plus auxiliary subunits conferring distinct electrophysiological kinetics, unique molecular structures, and different pharmacological sensitivities (*Jackson et al., 2011*; *Bowie, 2018*; *Azumaya et al., 2017*; *Twomey et al., 2018*). In the adult brain, most AMPAR tetramers contain an RNA-edited form of the GluA2 subunit that makes the channel relatively impermeable to $Ca^{2+}$, resulting in $Ca^{2+}$-impermeable AMPARs (CI-AMPARs; *Sommer et al., 1991*; *Higuchi et al., 1993*). AMPARs lacking edited GluA2 are called $Ca^{2+}$-permeable AMPARs (CP-AMPARs) because they have greater permeability to $Ca^{2+}$ and larger overall ionic conductance, carried mainly by $Na^+$ (*Hollmann et al., 1991*; *Geiger et al., 1995*). The expression of GluA2-lacking CP-AMPARs is downregulated in the developing brain (*Pickard et al., 2000*; *Kumar et al., 2002*; *Henley and Wilkinson, 2016*). However, CP-AMPARs persist or even increase with developmental maturation in some neurons of the auditory brainstem where CP-AMPARs enriched in GluA3 and GluA4 subunits are thought to be essential for fast transmission of acoustic signals (*Trussell, 1997*; *Gardner et al., 2001*; *Lawrence and Trussell, 2000*; *Sugden et al., 2002*; *Wang and Manis, 2005*; *Youssoufian et al., 2005*; *Lujan et al., 2019*).

Cochlear afferent projections process fast auditory signals through innervation of the anteroventral cochlear nucleus, at the endbulb of Held synapses onto bushy cells, where the AMPARs are comprised mainly of GluA3 and GluA4 subunits with high $Ca^{2+}$ permeability and rapid desensitization kinetics (*Wang et al., 1998*; *Rubio et al., 2017*). Mice lacking the GluA3 subunit have impaired auditory processing due to effects on synaptic transmission associated with altered ultrastructure of synapses between endbulbs and bushy cells (*García-Hernández et al., 2017*; *Antunes et al., 2020*). Mice lacking the GluA4 subunit have altered acoustic startle responses and impaired transmission at the next synaptic relay at the calyx of Held in the brainstem, a high-fidelity central synapse (*Yang et al., 2011*; *García-Hernández and Rubio, 2022*). The rapid processing of auditory signals in the brainstem is supported by high-fidelity initial encoding of sound at peripheral synapses between cochlear inner hair cells (IHCs) and spiral ganglion neurons (SGNs) (*Rutherford and Moser, 2016*; *Rutherford et al., 2021*), however, relatively little is known about how synapse ultrastructure, molecular composition, and overall abundance of cochlear AMPARs depends on specific pore-forming subunits.

In mice, the developmental onset of hearing function begins at the end of the second postnatal week, followed by activity-dependent maturation and neuronal diversification that depends on glutamatergic transmission in the SGNs (*Shrestha et al., 2018*; *Sun et al., 2018*; *Petitpré et al., 2018*; *Petitpré et al., 2022*). Heterogeneity of the SGNs and the ribbon synapses driving them results in auditory nerve fibers with mutually diverse sound response properties correlated to differences in synapse structure and position of innervation on the IHC modiolar–pillar axis (*Merchan-Perez and Liberman, 1996*; *Ohn et al., 2016*). Each primary auditory nerve fiber (i.e., type-I SGN) is unbranched and driven to fire spikes by the release of glutamate from an individual IHC-ribbon synapse driving a single, large postsynaptic density (PSD) of approximately 850 nm in length, on average (cat: *Liberman, 1980*; mouse *Payne et al., 2021*). In the mature cochlea, the PSD is populated with AMPARs comprised of subunits GluA2–4 but not GluA1 (*Niedzielski and Wenthold, 1995*; *Matsubara et al., 1996*; *Parks, 2000*; *Shrestha et al., 2018*). Afferent signaling in the auditory nerve, as well as noise-induced excitotoxicity at cochlear afferent synapses (a form of synaptopathy), depends on activation of AMPARs (*Ruel et al., 2000*; *Hu et al., 2020*). However, the dependence of cochlear AMPAR function and pathology on specific pore-forming subunits is unclear.

Here, we examined the influence of GluA3 subunits on afferent synapse ultrastructure and on AMPAR subunit molecular anatomy in the PSD of the auditory nerve fiber in the mouse cochlea, with attention to GluA2 and GluA4 *flip* and *flop* isoforms and to positions of SGN innervation on the IHC modiolar–pillar axis. At the central auditory nerve projection in the cochlear nucleus, at the endbulb of Held synapse, GluA3 is required for both post- and presynaptic maturation of synapse structure and function (*García-Hernández et al., 2017*; *Antunes et al., 2020*). Therefore, we also examined presynaptic ribbon morphology in relation to position on the IHC modiolar–pillar axis, which is expected to show smaller and more spherical ribbons on the side of the IHC facing the pillar cells and outer hair cells (pillar side) relative to the ribbons on the modiolar side facing the ganglion (*Merchan-Perez*

*and Liberman, 1996*; *Payne et al., 2021*). Our findings in young adult male GluA3[KO] mice include dysregulation of GluA2 and GluA4 subunit relative abundance and alterations in pre- and postsynaptic ultrastructure associated with an increased vulnerability to glutamatergic synaptopathy at ambient, background levels of sound. These structural and molecular alterations at the cochlear ribbon synapses of presymptomatic 5-week-old male GluA3[KO] mice appear to be pathological, preceding the reduction in ABR wave-1 amplitudes observed at 2 months of age (*García-Hernández et al., 2017*).

## Results

### Cochlear responses to sound and transcriptional splicing of *Gria2* and *Gria4* mRNA isoforms are similar in 5-week-old GluA3[WT] and GluA3[KO] mice

The four AMPAR pore-forming subunits GluA1–4 are encoded by four genes, *Gria1–4*. Here, we studied mice with normal or disrupted *Gria3* (i.e., GluA3[WT] or GluA3[KO]; *García-Hernández et al., 2017*; *Rubio et al., 2017*). We first determined whether the 5-week-old C57BL/6J GluA3[WT] and GluA3[KO] differed in cochlear responses to sound. Complete statistical details are included in source data files online. Our ABR analysis showed no differences between genotypes in clicks or pure tone thresholds or wave-1 amplitude or latency (*Figure 1A*). We note that male GluA3[KO] and GluA3[WT] mice at 2 months of age have similar ABR thresholds but GluA3[KO] mice have reduced ABR wave-1 amplitudes (*García-Hernández et al., 2017*), suggesting cochlear deafferentation between postnatal weeks 5 and 9.

We then asked if disruption of *Gria3* affected expression of GluA1, GluA2, or GluA4 protein subunits in the cochlear spiral ganglion (the auditory nerve fiber, SGN somata) or the cochlear nucleus (*Figure 1B, C*). In WT mice, mature SGNs express GluA2, GluA3, and GluA4 subunits of the AMPAR, but not GluA1 (*Niedzielski and Wenthold, 1995*; *Matsubara et al., 1996*; *Parks, 2000*; *Shrestha et al., 2018*). With immunolabeling, we observed GluA2 and GluA4 in the SGNs of both genotypes, and we confirmed that SGNs lacked GluA1 in GluA3[WT] mice, as expected. Moreover, we did not observe compensatory GluA1 expression in SGNs of GluA3[KO] (*Figure 1B*, left). We also checked immunolabeling of GluA1 on brainstem sections containing the ventral cochlear nucleus and cerebellum. As expected, we found GluA1 immunoreactivity in the cerebellar Bergmann glia of GluA3[WT] and GluA3[KO] mice (*Matsui et al., 2005*; *Douyard et al., 2007*). In contrast, the ventral cochlear nucleus of 5-week-old mice lacked GluA1 immunoreactivity in GluA3[WT] as previously shown (*Wang et al., 1998*). Also similar to the SGNs, we found no compensational expression of GluA1 in neurons of the cochlear nucleus from GluA3[KO] mice (*Figure 1B*, right). At ribbon synapses in the cochlea, the PSDs on the postsynaptic terminals of SGNs expressed GluA2, 3, and 4 in GluA3[WT] as previously shown (*Sebe et al., 2017*), while PSDs in GluA3[KO] lacked specific immunolabeling for GluA3 (*Figure 2*). This confirmed the deletion of GluA3 subunits was effective in SGNs of GluA3[KO] mice, as previously shown in the cochlear nucleus (*García-Hernández et al., 2017*; *Rubio et al., 2017*), and was not associated with compensatory upregulation of GluA1 subunits.

Two unique isoforms termed *flip* and *flop* are generated by alternative splicing of the mRNA encoding each of the pore-forming GluA subunits. In the brain, *flip* and *flop* splice variants are expressed in distinct but partly overlapping patterns and impart different desensitization kinetics (*Sommer et al., 1990*). The chicken and rat cochlear nuclei express predominantly the fast-desensitizing *flop* isoforms (*Schmid et al., 2001*; *Sugden et al., 2002*). With qRT-PCR, we determined whether *Gria3* disruption altered posttranscriptional *flip* and *flop* splicing of mRNA for GluA2 (*Gria2* gene) or GluA4 (*Gria4* gene) in the cochlea. We measured the levels of *flip* or *flop* for *Gria2* and *Gria4* mRNA and compared GluA3[WT] to GluA3[KO] (*Figure 1C*). Although GluA3[KO] exhibited increased variance in transcript abundance among samples for all four isoforms and a trend toward greater mRNA abundance in all four comparisons, we found no significant differences between GluA3[KO] and GluA3[WT]. In addition, we calculated the *flip/flop* ratios for *Gria2* and *Gria4* in GluA3[WT] and GluA3[KO] and found no differences between genotype (*Gria2 flip/flop* ratio WT: 0.67 ± 0.02, GluA3[KO]: 0.67 ± 0.01, p = 0.8; *Gria4 flip/flop* ratio WT: 0.70 ± 0.001, GluA3[KO]: 0.70 ± 0.01, p = 0.9 paired *t*-test two-tailed).

Thus, *Gria3* disruption did not affect hearing sensitivity at 5 weeks of age, in contrast to 8 weeks of age when ABR peak amplitudes were reduced (*García-Hernández et al., 2017*). Taken together with previous work, this suggests the 5-week-old GluA3[KO] cochlea may be in a pathological but presymptomatic, vulnerable state. The levels of *Gria2* or *Gria4 flip* or *flop* mRNA isoforms in cochleae of male

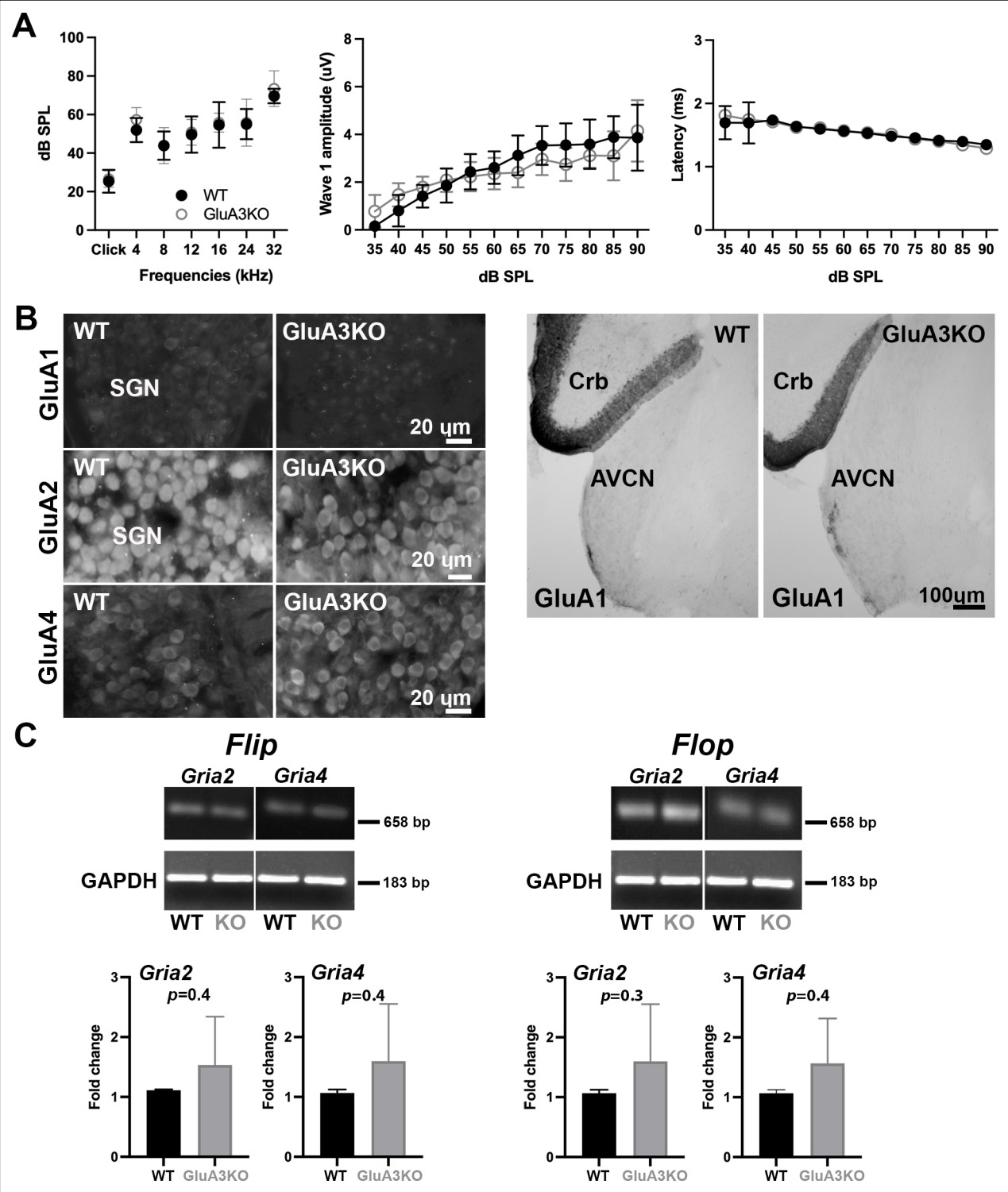

**Figure 1.** ABRs, GluA1 and GluA2 immunolabeling and qRT-PCR in GluA3^WT and GluA3^KO. (**A**) Mean ABR thresholds ( ± standard deviation [SD]) were similar between male GluA3^WT and GluA3^KO mice ($F_{(1, 168)}$ = 2.659, p = 0.11; two-way analysis of variance (ANOVA); GluA3^WT $n$ = 13; GluA3^KO $n$ = 13). In GluA3^WT and GluA3^KO, there was a main effect of sound frequency ($F_{(6, 168)}$ = 78.78, p < 0.0001). For ABR wave-1 amplitudes ( ± SD), there was an effect of sound ilevel ($F_{(11, 192)}$ = 49.62, p < 0.0001), but mean amplitudes were similar between genotypes ($F_{(1, 292)}$ = 2.458, p = 0.118; two-way ANOVA). For ABR wave-1 latencies ( ± SD), there was a main effect of sound level in both genotypes ($F_{(1, 288)}$ = 47.11, p < 0.0001) and mean latencies were similar between GluA3^WT and GluA3^KO mice ($F_{(1, 288)}$ = 0.1273, p = 0.7215; two-way ANOVA). (**B**) Micrographs show immunolabeling for GluA1, GluA2, and GluA4 on spiral ganglion neuron (SGN) somata, and for GluA1 on the anteroventral cochlear nucleus (AVCN) and cerebellum (Crb) of GluA3^WT and GluA3^KO mice.

*Figure 1 continued on next page*

*Figure 1 continued*

Immunolabeling for GluA2 and GluA4 is observed on SGNs of both genotypes. In contrast, immunolabeling for GluA1 was not observed on SGNs nor in the AVCN of GluA3$^{WT}$ or GluA3$^{KO}$ mice, but was observed in the cerebellar Bergmann glia of both genotypes. Scale bars: 20 and 100 μm. (**C**) Images of *Gria2* and *Gria4 flip* and *flop*, and GAPDH gels of GluA3$^{WT}$ and GluA3$^{KO}$ inner ears. Histograms show fold change ( ± SD) of qRT-PCR product. Paired *t*-test, two-tailed; bp: base pairs.

The online version of this article includes the following source data for figure 1:

**Source data 1.** Data and statistical analysis for the ABR, PCR gels and qRT-PCR for GluA3$^{WT}$ and GluA3$^{KO}$ mice.

**Source data 2.** Raw unedited PCR acrylamide gels for *Gria2* and *Gria4 flip/flop* in GluA3$^{WT}$ and GluA3$^{KO}$.

**Source data 3.** Figures of the uncropped PCR acrylamide gels for *Gria2* and *Gria4 flip/flop* in GluA3$^{WT}$, and GluA3$^{KO}$ with relevant bands and lanes clearly labeled.

mice at 5 weeks of age were similar in GluA3$^{WT}$ and GluA3$^{KO}$. In both genotypes, expression of *Gria2* and *Gria4 flop* isoforms appeared to exceed expression of the flip isoforms.

## Pre- and postsynaptic ultrastructural features of IHC-ribbon synapses are disrupted in the organ of Corti of GluA3$^{KO}$ mice

Given the similarity of cochlear responses to sound measured by ABR in male GluA3$^{WT}$ and GluA3$^{KO}$ mice at 5 weeks of age, we next asked if the ultrastructure of IHC-ribbon synapses was similar as well. Qualitatively, in GluA3$^{WT}$ and GluA3$^{KO}$ the general structure and cellular components of the sensory epithelia were similar to published data of C57BL/6 mice (not shown; *Ohlemiller and Gagnon, 2004*). Synapses from the mid-cochlea of both GluA3$^{WT}$ and GluA3$^{KO}$ mice had electron-dense pre- and post-synaptic membrane specializations and membrane-associated presynaptic ribbons (*Figures 3 and 4*).

## Ultrastructure in C57BL/6 GluA3$^{WT}$

A total of 29 synapses of GluA3$^{WT}$ mice were analyzed in three dimensions (3D) using serial sections (on average, 7 ultrathin sections per PSD). Of this total, 17 were on the modiolar side and 12 on the pillar side of the IHCs (*Figure 3A, B*, *Figure 3—figure supplement 1*). In our sample of the modiolar-side synapses, 11 had one single ribbon whereas 6 had two ribbons, so for the analysis of the PSD we classified the synapses as modiolar-1 and modiolar-2, for single and double ribbons, respectively. All the pillar-side synapses analyzed had a single ribbon. We then compared the PSD surface area and volume among the synapses of modiolar-1, modiolar-2, and pillar sides. One-way analysis of variance (ANOVA) comparison of the PSD surface area was significant (p = 0.007). Pairwise comparisons showed that the PSD surface areas were similar (p = 0.98) for single- and double-ribbon synapses of the modiolar side (modiolar-1 mean = 0.52 ± 0.15 μm$^2$; modiolar-2 mean = 0.56 ± 0.15 μm$^2$). However, in C57BL/6 WT mice, we observed that PSD surface area was larger for modiolar-side synapses compared to pillar-side synapses (p = 0.014 modiolar-1 vs. pillar, and p = 0.03 modiolar-2 vs. pillar; pillar mean: 0.40 ± 0.06 μm$^2$) (*Figure 3C*, left). We then measured PSD volumes, which were ~2× larger on the modiolar side, on average, but not significantly different (p = 0.051 one-way ANOVA), (modiolar-1 mean = 0.010 ± 0.006 μm$^3$; modiolar-2 mean = 0.008 ± 0.004 μm$^3$; pillar mean = 0.005 ± 0.002 μm$^3$) (*Figure 3C*, left). One-way ANOVA of the PSD linear length showed no significant differences among synapse type (p = 0.17; modiolar-1, *n* = 21, mean length: 666 ± 186 nm; modiolar-2, *n* = 6, mean length: 709 ± 234 nm; pillar, *n* = 16, mean length: 572 ± 135 nm) (*Figure 3D*, left). Overall, our analysis shows the PSD surface areas of modiolar-side synapses are significantly larger than those of the pillar side in GluA3$^{WT}$ on C57BL/6 background.

Presynaptic ribbon volume of GluA3$^{WT}$ was similar between modiolar- and pillar-side synapses (p = 0.57; modiolar mean: 0.0029 ± 0.001 μm$^3$; pillar mean: 0.0023 ± 0.0009 μm$^3$; Mann–Whitney *U*-test, two-tailed) (*Figure 3C*, right). In contrast, the surface area of modiolar-side ribbons was found significantly larger (p = 0.002; modiolar mean: 0.141 ± 0.123 μm$^2$; pillar mean: 0.074 ± 0.02 μm$^2$; Mann–Whitney *U*-test, two-tailed) (*Figure 3C*, right). For all pairwise statistical comparisons of ultrastructure in the following, we used the two-tailed Mann–Whitney *U*-test.

Analysis of the major axis and shape of synaptic ribbons in GluA3$^{WT}$ showed that the IHC synaptic ribbons on the modiolar side had longer major ribbon axes (p < 0.0001; mean: 274 ± 75 nm) and less circularity (p < 0.0001; mean: 0.51 ± 0.12) compared to the pillar-side ribbons (mean major axis: 180 ± 54 nm; mean circularity: 0.9 ± 0.06). These data show that ribbons on the modiolar side of GluA3$^{WT}$

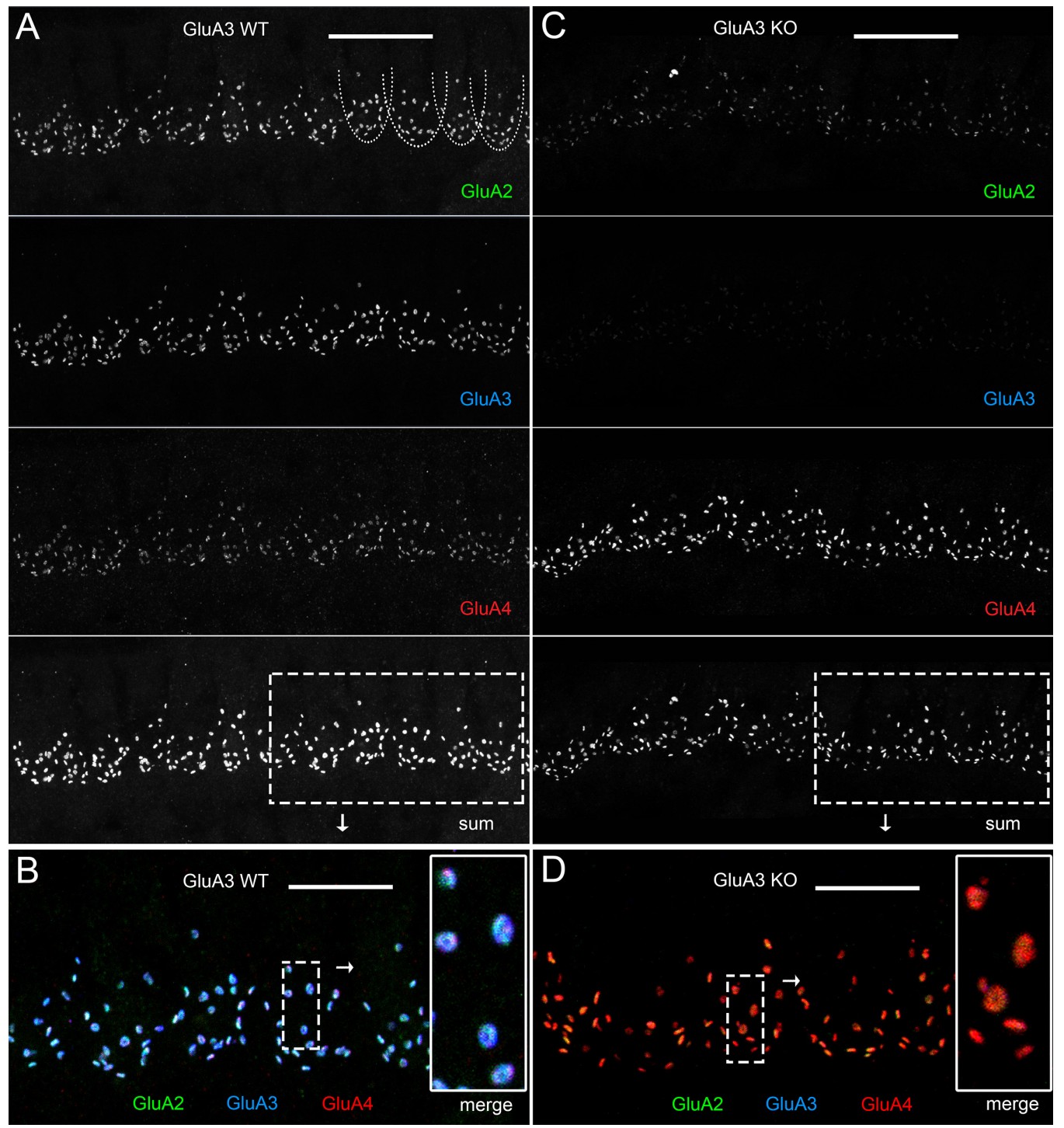

**Figure 2.** Immunohistofluorescence of α-amino-3-hydroxy-5-methyl-4-isoxazole propionic acid receptor (AMPAR) pore-forming subunits GluA2, 3, and 4 on spiral ganglion neuron postsynaptic terminals in the organ of Corti. Confocal microscope immunofluorescence images of afferent ribbon synapses in organ of Corti whole-mount samples from GluA3[WT] (left) and GluA3[KO] mice (right) in the mid-cochlea. Anti-GluA2 (green), -GluA3 (blue), and -GluA4 (red) labels the postsynaptic AMPAR subunits encoded by the *Gria2*, *Gria3*, and *Gria4* genes, respectively. Each subpanel displays synaptic puncta of approximately 12 inner hair cells (IHCs). Scale bars: 20 μm (**A, C**); 10 μm (**B, D**). (**A**) From top to bottom: GluA3[WT] in grayscale for anti-GluA2, 3, 4, and the sum of the three. In the GluA2 subpanel, the basolateral membranes of four IHCs are indicated by dashed curves. (**B**) Merged color image of the region of interest indicated by the dashed rectangle in panel A. Inset on right: enlargement of the dashed rectangular region of interest on left shows five postsynaptic AMPA receptor arrays of ribbon synapses from one IHC. (**C**) From top to bottom: GluA3[KO] in grayscale for anti-GluA2, 3, 4, and the sum of the three. (**D**) Merged color image of the region of interest indicated in panel C. Inset: enlargement of a rectangular region of interest shows several postsynaptic AMPAR arrays of ribbon synapses from one IHC.

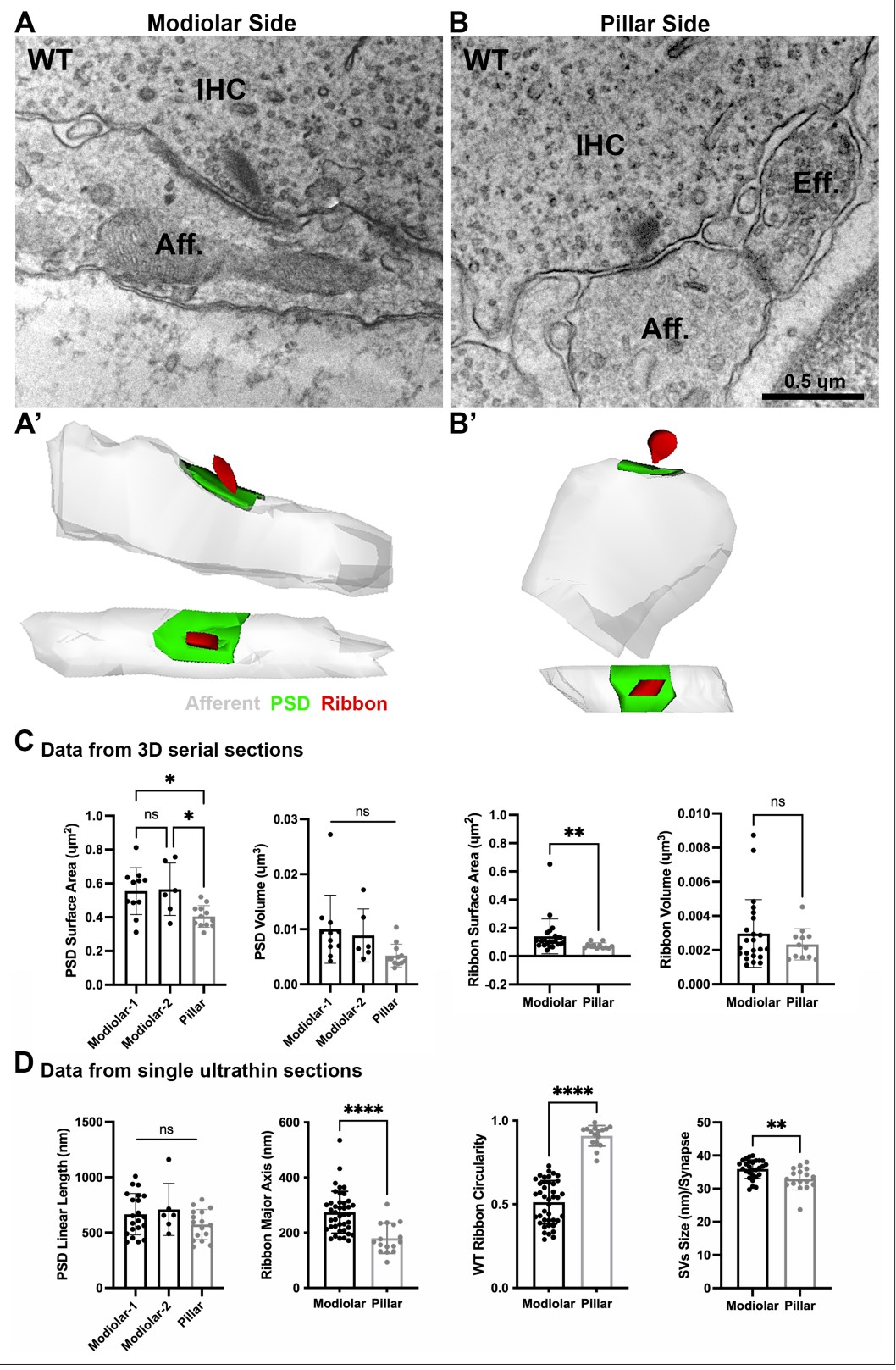

**Figure 3.** Ultrastructural features of GluA3[WT] IHC-ribbon mid-cochlear synapses. Transmission electron microscopy (TEM) micrographs of IHC synapses on the modiolar (**A**) and pillar sides (**B**). Aff.: afferent; IHC: inner hair cell; Eff.: efferent terminal. Scale bar: 0.5 µm. (**A′, B′**) Three-dimensional (3D) reconstructions of the IHC-ribbon synapses are shown in A and B. Representative serial electron micrograph images of modiolar- and pillar-side ribbon synapses

*Figure 3 continued on next page*

*Figure 3 continued*

are shown in *Figure 3—figure supplement 1*. (**C**) Plots of the quantitative data of the surface area, and volume of the postsynaptic densities (PSDs) and ribbons obtained from the 3D reconstructions of GluA3$^{WT}$ mice. The error bar corresponds to ± standard deviation (SD). (**D**) Plots of the quantitative data from single ultrathin sections of the linear length of the PSD, major axis, and circularity of the ribbons, and the average size of synaptic vesicles (SVs)/synapse of GluA3$^{WT}$ mice. The error bar corresponds to ± SD; one-way Anova * p < 0.05, ns: not significant; Mann-Whitney two-tailed U-test, ** p < 0.01, *** p < 0.0001, ns: not significant.

The online version of this article includes the following source data and figure supplement(s) for figure 3:

**Source data 1.** Data and statistical analysis for the ultrastructural analysis of GluA3$^{WT}$ mice.

**Figure supplement 1.** Representative serial electron micrographs and corresponding three-dimensional (3D) reconstructions of modiolar- or pillar-side inner hair cell (IHC)-ribbon synapses of the GluA3$^{WT}$ mice.

---

IHCs are elongated, while those on the pillar side are more round in shape (*Figure 3D*, center), as previously shown for C57BL/6 mice at 5 weeks of age (*Payne et al., 2021*). Analysis of synaptic vesicle (SV) size showed that the SVs of modiolar-side synapses were larger (p = 0.0029) than those of the pillar-side synapses (modiolar mean: 36 ± 3 nm; pillar mean: 33 ± 4 nm; *Figure 3D*, right). In summary, GluA3$^{WT}$ synapses of the modiolar side had larger PSD surface areas, more elongated and less circular ribbons with greater surface area, and larger SVs compared with synapses of the pillar side.

## Ultrastructure in C57BL/6 GluA3$^{KO}$

From GluA3$^{KO}$, a total of 26 synapses were analyzed in 3D with serial sections (on average, 7 ultrathin sections per PSD). Of this total, 16 were on the modiolar side and 10 on the pillar side of the IHCs (*Figure 4A, B*, *Figure 4—figure supplement 1*). As with synapses from GluA3$^{WT}$, for the analysis of the PSDs of GluA3$^{KO}$ we classified the modiolar-side synapses as modiolar-1 (single ribbon; *n* = 11) or modiolar-2 (double ribbons; *n* = 5). In contrast to the pillar-side synapses of GluA3$^{WT}$ that had only single ribbons, we found two pillar-side synapses of GluA3$^{KO}$ cochleae with double ribbons (e.g., *Figure 4—figure supplement 1*). These pillar-side synapses with double ribbons were not included in our analysis. We then compared the PSD surface areas and volumes among synapses on the modiolar (modiolar-1 and modiolar-2) and pillar sides. One-way ANOVA comparison of the PSD surface area was not significant (p = 0.67; modiolar-1 mean: 0.51 ± 0.18 µm$^2$; modiolar-2 mean: 0.55 ± 0.14 µm$^2$; pillar mean: 0.57 ± 0.16 µm$^2$). One-way ANOVA analysis of the PSD volume was not significant (p = 0.65; modiolar-1 mean: 0.0085 ± 0.004 µm$^3$; modiolar-2 mean: 0.0068 ± 0.001 µm$^3$; pillar mean: 0.0083 ± 0.003 µm$^3$) (*Figure 4C*, left). The PSD linear lengths were not significantly different (p = 0.07, one-way ANOVA; modiolar-1, *n* = 30, mean length: 629 ± 147 nm; modiolar-2, *n* = 6, mean length: 543 ± 32 nm; pillar, *n* = 17, mean length: 684 ± 118 nm) (*Figure 4D*, left).

Analysis of GluA3$^{KO}$ presynaptic ribbon volumes showed that pillar-side synapses (mean: 0.0042 ± 0.001 µm$^3$) had larger volumes than modiolar-side synapses (mean: 0.0032 ± 0.001 µm$^3$; p = 0.047) (*Figure 4C*, right), in contrast to GluA3$^{WT}$. Also, unlike GluA3$^{WT}$, the surface area was similar between ribbons on the modiolar and pillar sides of GluA3$^{KO}$ (modiolar mean: 0.14 ± 0.14 µm$^2$, pillar mean: 0.17 ± 0.25 µm$^2$; p = 0.91) (*Figure 4C*, right).

The major ribbon axes from GluA3$^{KO}$ were similar on the modiolar side (mean: 199 ± 65 nm) and pillar side (mean: 201 ± 89 nm; p = 0.9). Modiolar-side ribbons had less circularity than pillar-side ribbons (modiolar mean: 0.75 ± 0.10; pillar mean: 0.85 ± 0.07; p = 0.01) (*Figure 4D*, right), but this difference was lesser than the difference observed in GluA3$^{WT}$. Opposite to the pattern in GluA3$^{WT}$, SVs of modiolar-side synapses were smaller than those of the pillar side in GluA3$^{KO}$ (modiolar: 35 ± 5 nm; pillar: 38 ± 3 nm; p = 0.04) (*Figure 4D*, right). In summary, unlike GluA3$^{WT}$, GluA3$^{KO}$ synapses of the modiolar side had similar PSD and ribbon surface areas and ribbon long axes as pillar-side synapses, and smaller SVs, demonstrating disruption of modiolar–pillar synaptic differentiation during development.

## IHC modiolar–pillar differences are eliminated or reversed in GluA3$^{KO}$

We then compared PSDs and ribbons among GluA3$^{WT}$ and GluA3$^{KO}$ mice on the modiolar and pillar sides (*Figure 5A*). Overall, the PSD surface area and volume of the modiolar-side synapses (modiolar-1 and modiolar-2) were similar between genotypes (surface area: p = 0.85; volume: p = 0.62; one-way ANOVA). In contrast, the mean surface area and volume of the pillar-side PSDs were larger in

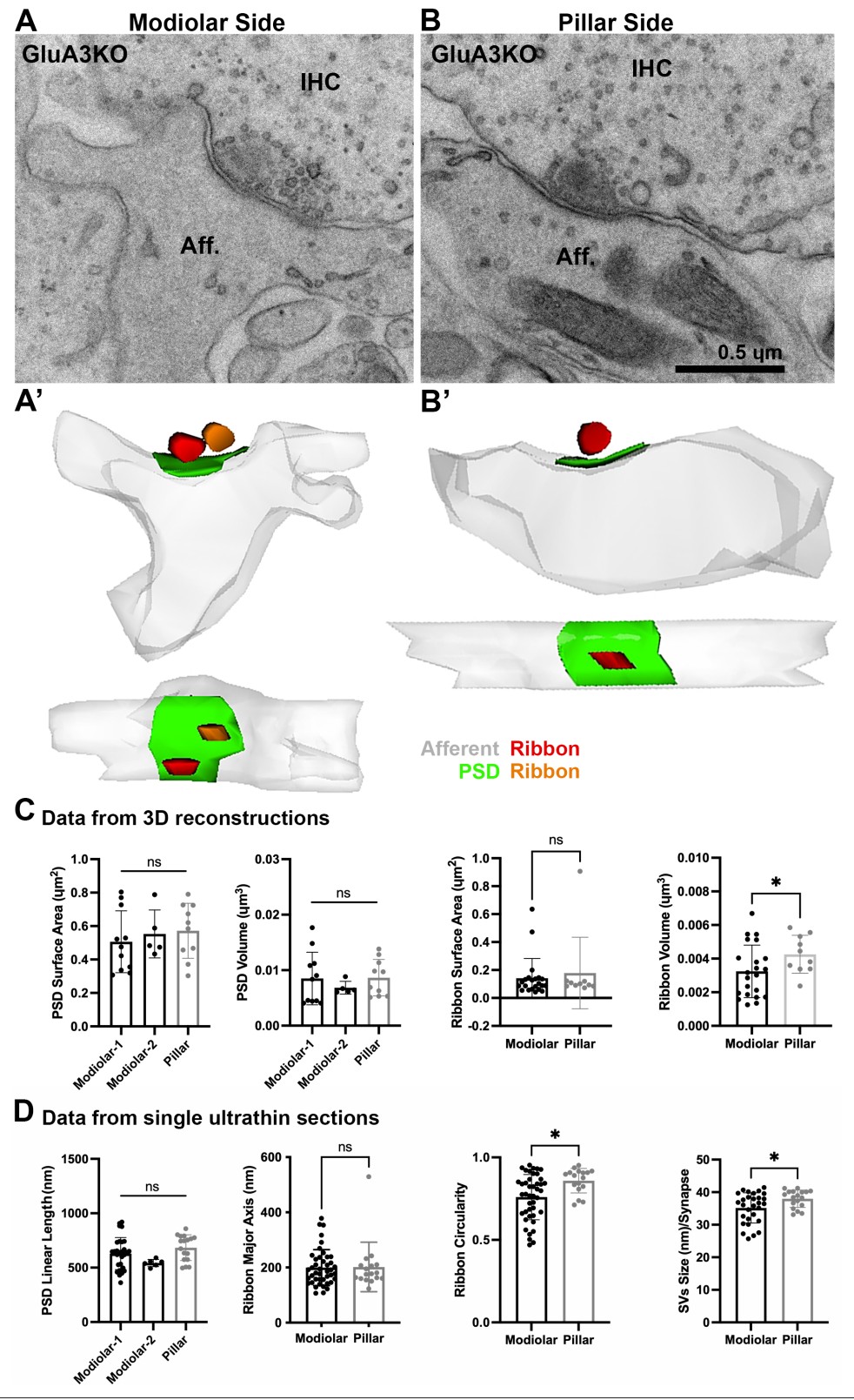

**Figure 4.** Ultrastructural features of GluA3[KO] IHC-ribbon mid-cochlear synapses. Transmission electron microscopy (TEM) micrographs of IHC synapses on the modiolar (**A**) and pillar sides (**B**) of GluA3[KO] mice. Aff.: afferent; IHC: inner hair cell. Scale bar: 0.5 μm. (**A', B'**) Three-dimensional reconstructions of the IHC-ribbon synapses are shown in A and B. Representative serial electron micrograph images of modiolar- and pillar-side ribbon synapses are

*Figure 4 continued on next page*

*Figure 4 continued*

shown in *Figure 4—figure supplement 1*. (**C**) Plots of the quantitative data of the surface area and volume of the postsynaptic densities (PSDs) and ribbons obtained from the 3D reconstructions of GluA3$^{KO}$ mice. The error bar corresponds to ± standard deviation (SD). (**D**) Plots of the quantitative data from single ultrathin sections of the linear length of the PSD, major axis and circularity of the ribbons and the average size of synaptic vesicles (SVs)/ synapse of GluA3$^{KO}$ mice. The error bar corresponds to ± SD; one-way ANOVA, ns: not significant; Mann-Whitney two-tailes U-test, * p < 0.05, ns: not significant.

The online version of this article includes the following source data and figure supplement(s) for figure 4:

**Source data 1.** Data and statistical analysis for the ultrastructural analysis of GluA3$^{KO}$ mice.

**Figure supplement 1.** Representative serial electron micrographs and corresponding three-dimensional (3D) reconstructions of modiolar- or pillar-side inner hair cell (IHC)-ribbon synapses of the GluA3$^{KO}$ mice.

GluA3$^{KO}$ than GluA3$^{WT}$ (surface area: p = 0.013; volume: p = 0.007) (*Figure 5A*, top). The average PSD length was similar between genotypes for the modiolar-side synapses (p = 0.29, one-way ANOVA). In contrast, as with surface area and volume for pillar-side synapses, the mean PSD length of GluA3$^{KO}$ pillar-side synapses was larger than GluA3$^{WT}$ (p = 0.02) (*Figure 5B*, top).

Synaptic ribbon volumes differed among modiolar and pillar groups of GluA3$^{WT}$ and GluA3$^{KO}$ (p = 0.04, one-way ANOVA). Pairwise comparisons showed that ribbon volumes of modiolar-side synapses were similar between GluA3$^{WT}$ and GluA3$^{KO}$ (p = 0.93). In contrast, the pillar-side synapses were larger in GluA3$^{KO}$ than in GluA3$^{WT}$ (p = 0.03) (*Figure 5A*, bottom right). One-way ANOVA analysis of the ribbon surface area between modiolar- and pillar-side synapses was similar between GluA3$^{WT}$ and GluA3$^{KO}$ (p = 0.39) (*Figure 5A*, bottom left).

Differences between the ribbon major axis were found between GluA3$^{WT}$ and GluA3$^{KO}$ (p < 0.0001, one-way ANOVA). On the modiolar side, analysis of the ribbon major axis length showed that those of the GluA3$^{KO}$ were significantly smaller than GluA3$^{WT}$ (p < 0.0001), whereas pillar-side synapses were similar in major axis length (p = 0.82) (*Figure 5B*, bottom left). Differences in ribbon circularity were also found between genotypes (p < 0.0001, one-way ANOVA). Paired comparisons showed that modiolar-side ribbons were significantly less circular in GluA3$^{WT}$ (p < 0.0001), whereas pillar-side ribbons were of similar circularity among genotypes (p = 0.62) (*Figure 5B*, bottom center). SVs size differed between genotypes (p = 0.0008, one-way ANOVA). Data showed that SVs of modiolar-side synapses were similar among genotypes (p = 0.84), while those of pillar-side synapses were significantly larger in GluA3$^{KO}$ (p = 0.0004) (*Figure 5D*, bottom right).

Altogether, our data of 5-week-old male mice show the AMPAR subunit GluA3 is essential to establish and/or maintain the morphological gradients of pre- and postsynaptic structures along the modiolar–pillar axis of the IHC. In GluA3$^{KO}$ enlargement of PSD surface area, presynaptic ribbon size, and SV size on the pillar side eliminated the modiolar–pillar morphological distinctions seen in 5-week-old male GluA3$^{WT}$ mice on C57BL/6 background. Next, we asked how these early ultrastructural changes in GluA3$^{KO}$ correlated with the number of ribbon synapses per IHC and the relative expression of GluA subunits at those synapses.

## An increase in GluA2-lacking synapses precedes a reduction in cochlear output in GluA3$^{KO}$ mice

Although GluA3$^{KO}$ mice had reduced ABR wave-1 amplitudes at 2 months of age (*García-Hernández et al., 2017*), their wave-1 amplitudes were not yet different from GluA3$^{KO}$ mice at 5 weeks of age (*Figure 1A*). Given the alterations in ribbon synapse ultrastructure at 5 weeks (*Figures 3–5*), we asked if synapse molecular anatomy was also affected in GluA3$^{KO}$ mice. Using confocal images of immunolabeled cochlear wholemounts, we analyzed the expression of CtBP2, GluA2, GluA3, and GluA4 at the ribbon synapses between IHCs and SGNs. Visual inspection of the images revealed robust anti-GluA3 labeling in GluA3$^{WT}$ and the absence of specific anti-GluA3 labeling in GluA3$^{KO}$. As well, there was an obvious reduction in GluA2 labeling and increase in GluA4 labeling in GluA3$^{KO}$ relative to GluA3$^{WT}$ (*Figure 2* and *Figure 6A*). Despite this, the numbers of paired synapses (CtBP2 + GluA2 + GluA4) per IHC were similar in the whole cochlea (*Figure 6B*; WT: 18.1 ± 2.8; KO: 17.3 ± 3.8; p = 0.94, Mann–Whitney two-tailed *U*-test) and within apical, middle, and basal cochlear regions (see *Figure 6* caption for more statistical details, Mann–Whitney two-tailed *U*-test unless otherwise noted). The numbers of ribbonless synapses per IHC (GluA2 + GluA4 – WT: 1.2 ± 0.64; KO: 1.1 ±

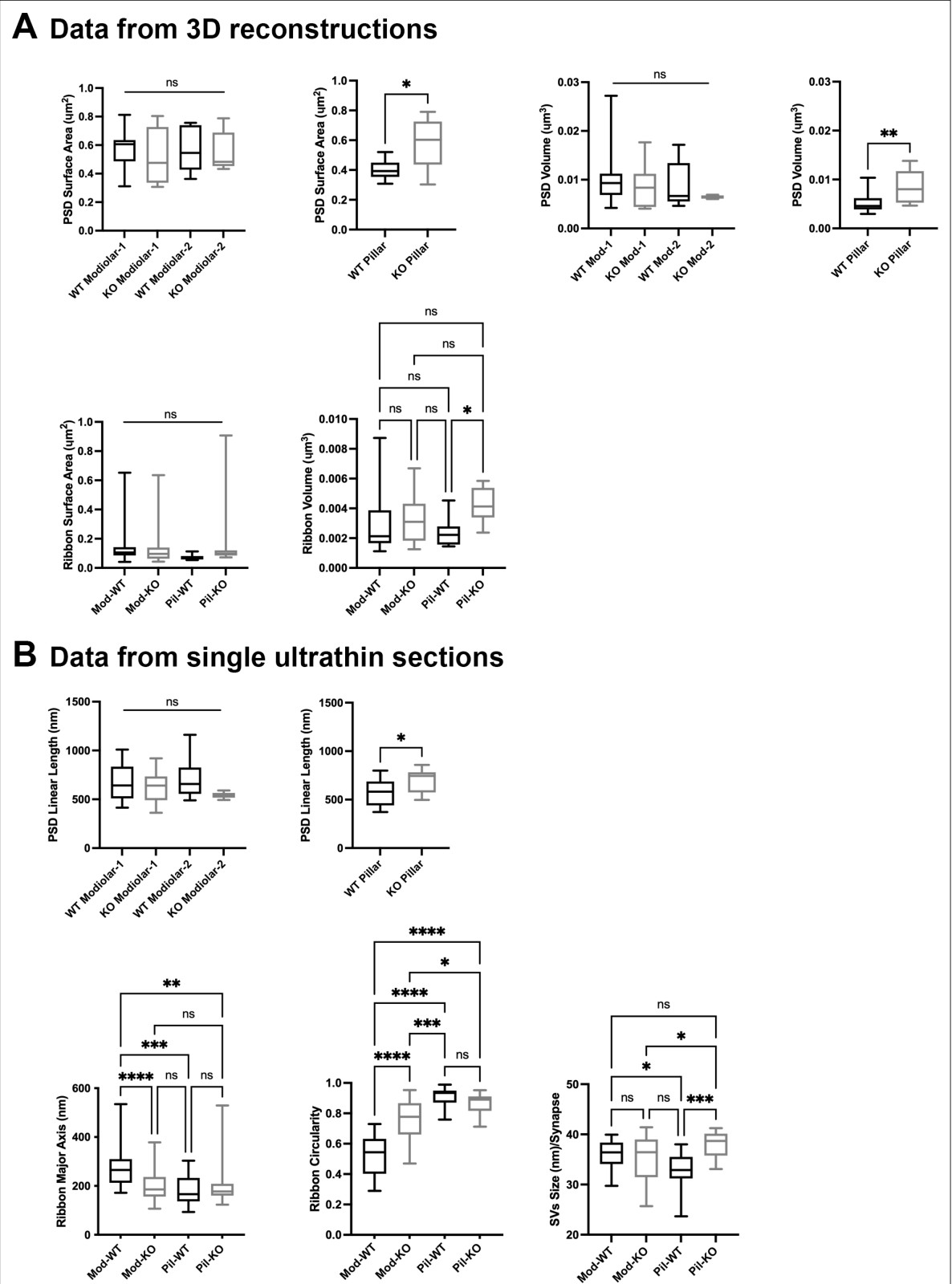

**Figure 5.** Inner hair cell (IHC) modiolar–pillar structural differences in presynaptic ribbon size, ribbon shape, and vesicle size seen in GluA3$^{WT}$ were diminished or reversed in GluA 3$^{KO}$. (**A**) Whisker plots show the quantitative data of the surface area and volume of the postsynaptic density (PSD) and ribbon volume of GluA3$^{WT}$ (black) and GluA3$^{KO}$ (gray) mice. The error bar corresponds to ± standard deviation (SD). (**B**) Whisker plots of the linear length of the PSD, major axis, and circularity of the ribbons of GluA3$^{WT}$ (black) and GluA3$^{KO}$ (gray) mice. Column histogram of the size of synaptic vesicles (SVs)

*Figure 5 continued on next page*

*Figure 5 continued*

of GluA3^WT (black) and GluA3^KO (gray). The error bar corresponds to ± SD; one-way ANOVA, * p < 0.05, ** p < 0.01, *** p < 0.005, p < 0.0001, ns: not significant; Mann-Whitney two-tailed U-test, * p < 0.05, ** p < 0.001.

The online version of this article includes the following source data for figure 5:

**Source data 1.** Data and statistical analysis for the comparison of the ultrastructural analysis of WT vs. GluA3^KO mice.

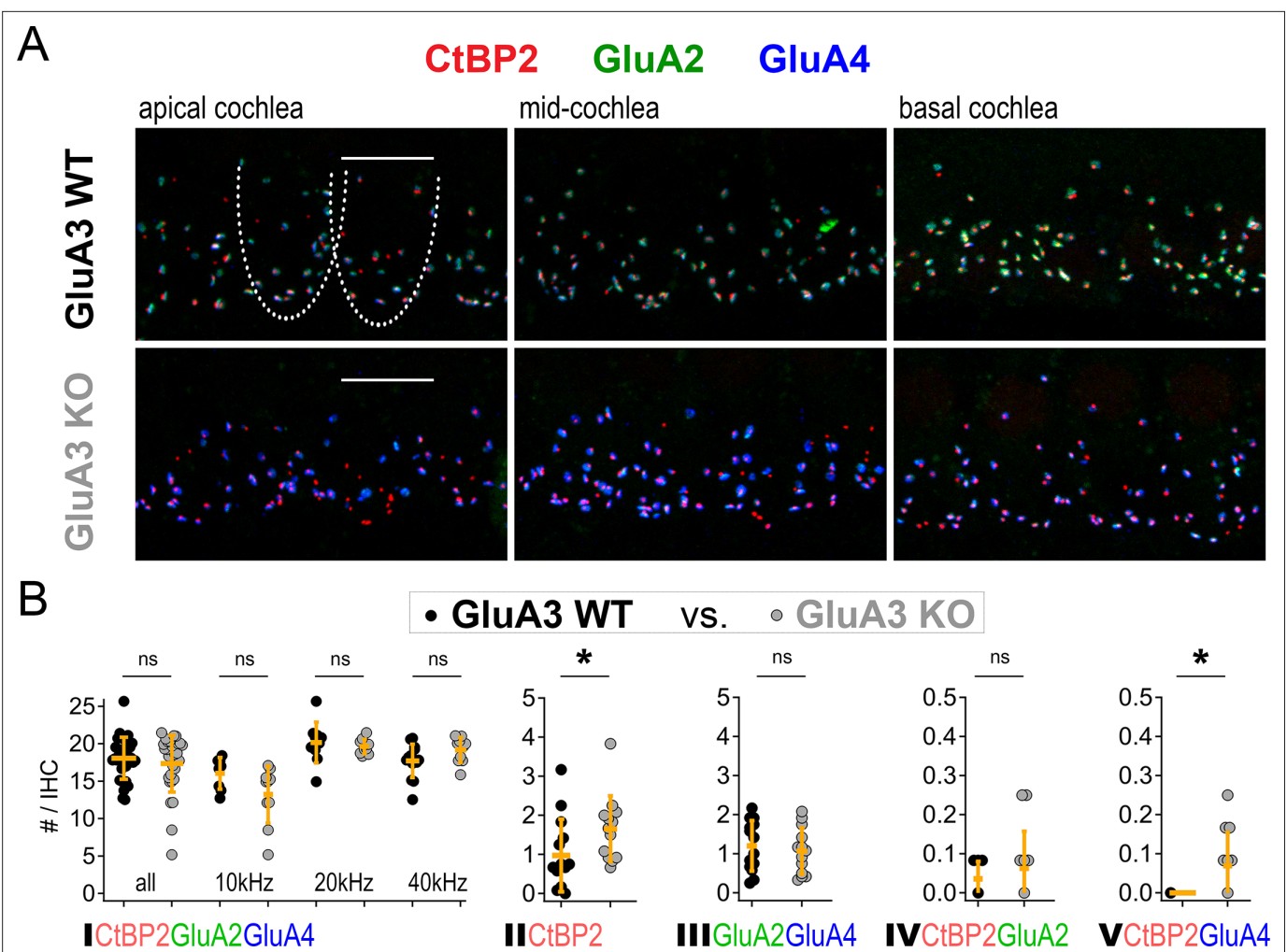

**Figure 6.** Inner hair cell (IHC)-ribbon synapse counts in 5-week-old male GluA3^WT and GluA3^KO mice. (**A**) Confocal microscope immunofluorescence images of afferent ribbon synapses in organ of Corti whole-mount samples from GluA3^WT (upper) and GluA3^KO mice (lower) in the apical, middle, and basal cochlea (left, middle, and right). Anti-CtBP2 labels the Ribeye protein in presynaptic ribbons (red); Anti-GluA2 labels the postsynaptic α-amino-3-hydroxy-5-methyl-4-isoxazole propionic acid receptor (AMPAR) subunit encoded by the *Gria2* gene (green); Anti-GluA4 labels the AMPAR subunit encoded by *Gria4* (blue). Each subpanel displays synaptic puncta of approximately four IHCs. Scale bar: 10 µm. (**B**) Quantification of ribbon synapse numbers in images from GluA3^WT (black: 2990 synapses; *n* = 32 images; 5 mice) and GluA3^KO (gray: *n* = 2814 synapses; *n* = 30 images; 5 mice). Each point represents the mean number of synapses per IHC per image; approximately 12 IHCs per image and 6 images per cochlea. (I) Paired synapses per IHC were similar in number for the whole cochlea (p = 0.94, *U*: 484, $n_{WT}$ = 32, $n_{KO}$ = 30) and in each of three tonotopic regions centered at 10 kHz (p = 0.08, *U*: 59, $n_{WT}$ = 8, $n_{KO}$ = 10), 20 kHz (p = 0.41, *U*: 61, $n_{WT}$ = 10, $n_{KO}$ = 10), or 40 kHz (p = 0.10, *U*: 42, $n_{WT}$ = 14, $n_{KO}$ = 10). (II) Lone or 'orphaned' ribbons (CtBP2-only) were significantly more frequent in GluA3^KO (p = 0.021, *U*: 44, $n_{WT}$ = 14, $n_{KO}$ = 13). (III) Ribbonless synapses (GluA2 + GluA4) were similar in number (p = 0.67, *U*: 100, $n_{WT}$ = 14, $n_{KO}$ = 13). (IV) Paired synapses lacking GluA4 (CtBP2 + GluA2) were similar in number (p = 0.81, *U*: 39, $n_{WT}$ = 7, $n_{KO}$ = 12). (V) Paired synapses lacking GluA2 (CtBP2 + GluA4) were observed in GluA3^KO (p = 0.028, *U*: 21, $n_{WT}$ = 7, $n_{KO}$ = 12) but not in GluA3^WT. Mann-Whitney two-tailed U-test; * p < 0.05, ns: not significant.

The online version of this article includes the following source data for figure 6:

**Source data 1.** Confocal data and statistics of synapse counts for GluA3^WT and GluA3^KO.

0.59; p = 0.67) and GluA4-lacking synapses (GluA2 + CtBP2 – WT: 0.035 ± 0.044; KO: 0.062 ± 0.095; p = 0.81) were not significantly different. In contrast, the numbers of lone ribbons per IHC (CtBP2-only – WT: 0.97 ± 0.92; KO: 1.6 ± 0.84; p = 0.021) and GluA2-lacking synapses per IHC (GluA4 + CtBP2 – WT: 0.0 ± 0.0; KO: 0.07 ± 0.09; p = 0.028) were significantly increased in GluA3$^{KO}$ relative to GluA3$^{WT}$ (*Figure 6B*).

## Loss of GluA3 expression reduces synaptic GluA2 and increases synaptic GluA4 subunits

Grayscale and color images (*Figures 2 and 6A*) revealed obvious reduction in GluA2 and increase in GluA4 subunit immunofluorescence per synapse, on average, as quantified in *Figure 7A* (*n* = 3 mid-cochlear images per genotype). The overall GluA fluorescence per synapse (GluA$_{Sum}$ = GluA2 + GluA3 + GluA4) tended to be smaller in GluA3$^{KO}$ due to the absence of GluA3 subunit fluorescence. Analysis of GluA2 and GluA4 puncta volumes and intensities in one exemplar mid-cochlear image from GluA3$^{WT}$ and GluA3$^{KO}$ mice revealed that ribbon synapses of GluA3$^{KO}$ mice had more compact AMPAR arrays (*Figure 7B*) with reduced GluA2 and increased GluA4 fluorescence intensity relative to GluA3$^{WT}$ (*Figure 7C*). The sublinear Volume vs. Intensity relationship for each GluA subunit suggests synaptic AMPAR density increases with the size of the GluA array in both genotypes (*Figure 7D*). Data from three mid-cochlear image stacks from each genotype are summarized in *Figure 7E–H* (same images as panel A). Relative to the mean of the summed pixel intensities per synapse in GluA3$^{WT}$ mice, the overall fluorescence of GluA subunits (GluA$_{Sum}$ = GluA2 + GluA3 + GluA4) was reduced in GluA3$^{KO}$ mice due to the absence of GluA3, despite the much larger increase in GluA4 fluorescence intensity relative to the reduction in GluA2 (*Figure 7E, F*). Relative to the mean GluA puncta volume per GluA3$^{WT}$ synapse, the mean volumes of GluA2, GluA4, and GluA$_{Sum}$ were all reduced in GluA3$^{KO}$ (*Figure 7G*). When normalized to the mean puncta volume per image in either group, the distributions of synapse volumes were broadened for GluA2 and GluA4 subunits in GluA3$^{KO}$ relative to GluA3$^{WT}$ (*Figure 7H*). For each image, we calculated the coefficient of variation (CV = SD/mean) in puncta volume for comparison by genotype. The volume of GluA$_{Sum}$ had a CV (mean ± SD, *n* = 3 images per genotype) of 0.38 ± 0.03 in GluA3$^{WT}$ vs. 0.51 ± 0.02 in GluA3$^{KO}$. For GluA2 volumes, the CVs were 0.39 ± 0.03 in GluA3$^{WT}$ vs. 0.65 ± 0.065 in GluA3$^{KO}$. For GluA4 volumes, the CVs were 0.38 ± 0.04 in GluA3$^{WT}$ vs. 0.51 ± 0.02 in GluA3$^{KO}$. Summed pixel intensity per synapse was more variable than volume and, again, more variable in GluA3$^{KO}$ than in GluA3$^{WT}$. For GluA$_{Sum}$ intensity, the CVs were 0.44 ± 0.04 in GluA3$^{WT}$ vs. 0.63 ± 0.02 in GluA3$^{KO}$. For GluA2 intensity, the CVs were 0.46 ± 0.03 in GluA3$^{WT}$ vs. 0.76 ± 0.04 in GluA3$^{KO}$. For GluA4 intensity, the CVs were 0.45 ± 0.05 in GluA3$^{WT}$ vs. 0.63 ± 0.02 in GluA3$^{KO}$.

To test the statistical significance and to confirm the differences observed in *Figure 7* in a larger data set from a replication cohort, we next assessed mean synaptic CtBP2, GluA2, and GluA4 volume and intensity per image in 14 image stacks from each genotype. In image stacks of sufficient quality, we also measured synapse position on the IHC modiolar–pillar axis to sort synapses into modiolar and pillar groups, dividing them at the midpoint of the range of ribbon centroids in the image *Z*-axis (*Figure 8—figure supplement 1*). Image means and group means are displayed in *Figure 8*. The volumes of CtBP2, GluA2, and GluA4 puncta were significantly smaller in GluA3$^{KO}$ relative to GluA3$^{WT}$ (*Figure 8A*, all, in µm³; CtBP2–GluA3$^{WT}$: 0.14 ± 0.02; CtBP2–GluA3$^{KO}$: 0.12 ± 0.01, p = 0.008; GluA2–GluA3$^{WT}$: 0.47 ± 0.06; GluA2–GluA3$^{KO}$: 0.39 ± 0.07, p = 0.0001; GluA4–GluA3$^{WT}$: 0.45 ± 0.05; GluA4–GluA3$^{KO}$: 0.36 ± 0.02, p = 4.9e$^{−6}$). When comparing modiolar- and pillar-side synapses, in both genotypes CtBP2 puncta tended to be larger on the modiolar side than the pillar side, on average, but the differences were not significant (p = 0.08 for GluA3$^{WT}$ and GluA3$^{KO}$). In GluA3$^{WT}$, GluA2 and GluA4 modiolar- and pillar-side puncta were not significantly different in volume (p = 0.42 GluA2; p = 0.23 GluA4). In contrast, in GluA3$^{KO}$, GluA2 and GluA4 puncta were significantly larger on the modiolar side than the pillar side (p = 0.001 GluA2; p = 0.0001 GluA4).

When comparing the two genotypes on either the modiolar sides or the pillar sides (*Figure 8A* and M, P), the group mean volumes tended to be larger in GluA3$^{WT}$ than GluA3$^{KO}$ for CtBP2, GluA2, and GluA4. However, the CtBP2 volumes were not significantly different when comparing genotypes in the modiolar-side or pillar-side anatomical subgroups (p = 0.098 for modiolar; p = 0.57 for pillar). For GluA2, only the pillar-side groups were significantly different, with a marked reduction in GluA3$^{KO}$ (0.36 ± 0.09 µm³) relative to GluA3$^{WT}$ (0.45 ± 0.06 µm³; p = 0.008). For GluA4, both the modiolar- and

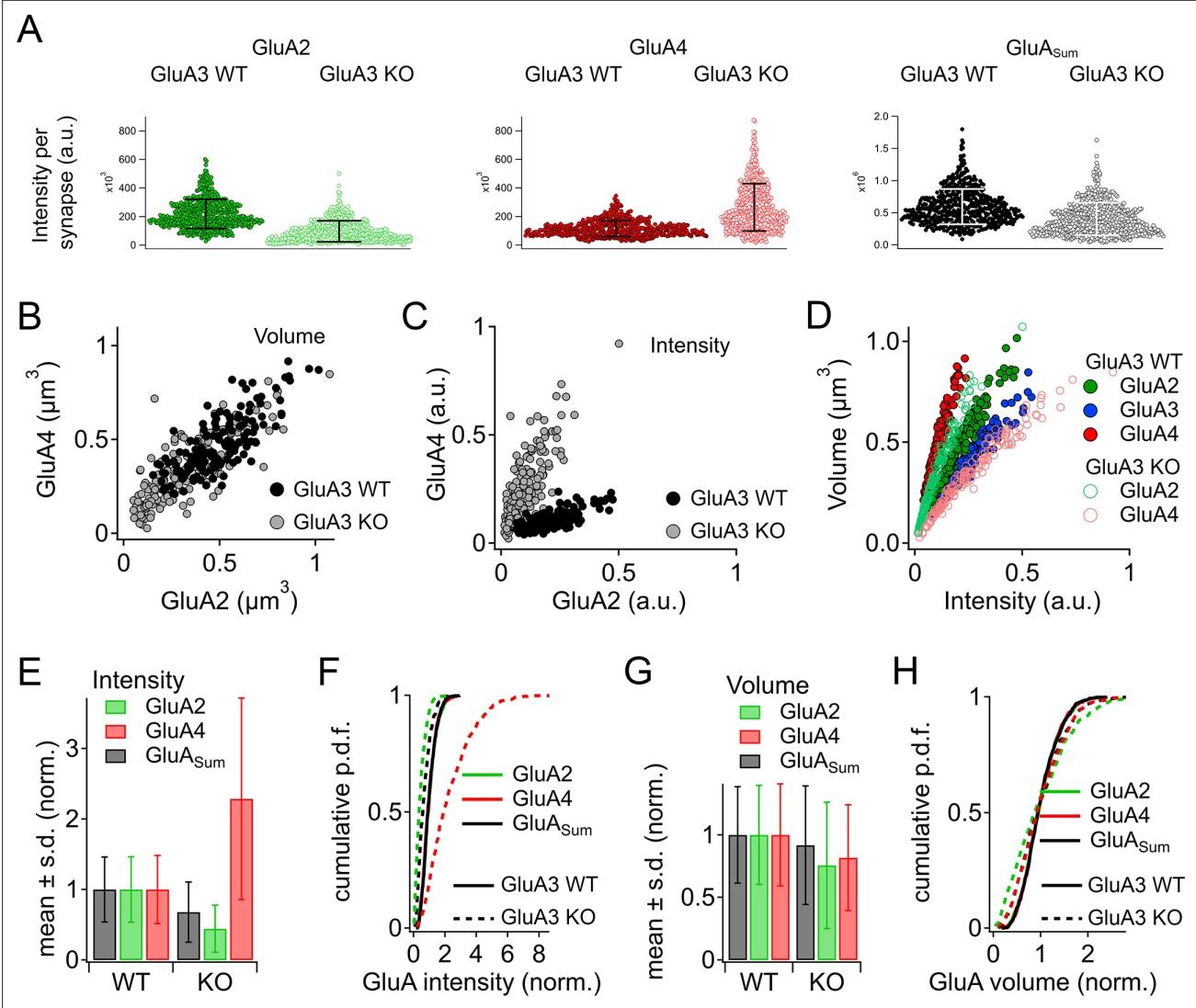

**Figure 7.** Alteration of α-amino-3-hydroxy-5-methyl-4-isoxazole propionic acid receptor (AMPAR) subunit expression in GluA3[KO] mice. (**A**) From images like in *Figure 2*: Summed pixel intensity per synapse (raw values, a.u.) for GluA2 (green), GluA4 (blue), and GluA_Sum (black). In each subpanel, GluA3[WT] is on left and GluA3[KO] is on right. Bars show mean ± standard deviation (SD); $n$ = 3 mid-cochlear images per genotype, assessed further in panels **E–H**. (**B**) Volume analysis of two exemplar images showing GluA4 vs. GluA2 volume per synapse (μm³) in the mid-cochlea of GluA3[WT] (black, $n$ = 148 synapses) and GluA3[KO] (gray, $n$ = 166 synapses). The distribution of GluA4 and GluA2 puncta are shifted to smaller volumes in GluA3[KO], although the upper ranges are unchanged. (**C**) Intensity analysis of the synapses in panel **B** (summed pixel intensity per synapse) reveals an increase in GluA4 and decrease in GluA2 immunofluorescence in GluA3[KO]. Intensity values were normalized to the maximum synapse intensity for GluA4. (**D**) Volume (μm³) vs. summed pixel intensity (norm.) per synapse for GluA3[WT] (filled circles) and GluA3[KO] (open circles) for GluA2, 3, and 4 puncta (green, blue, and red). The positive correlation is slightly sublinear. (**E**) Intensity analysis (sum of pixel intensities per synapse) of postsynaptic puncta grouped from GluA3[WT] ($n$ = 545 synapses from 3 images) or GluA3[KO] cochlea ($n$ = 513 synapses from 3 images) shows reduction of overall GluA intensity (GluA_Sum = GluA2 + GluA3 + GluA4, gray) and reduction in GluA2 intensity (green) with increase in GluA4 intensity (red) in GluA3[KO]. Data are normalized to the mean WT synapse intensity per group for GluA2, GluA4, or GluA_Sum. (**F**) Normalized data as in panel E displayed as cumulative distributions for GluA3[WT] (solid line) and GluA3[KO] (dashed lines). The overall intensity in GluA3[KO] (black dashed line, GluA_Sum) is reduced relative to GluA3[WT] (solid black line) due to lack of GluA3 and reduction in GluA2 (green) despite the relatively large increase in GluA4 (red). (**G**) GluA puncta volume analysis reveals a reduction of GluA2 and GluA4 volume per synapse in GluA3[KO] relative to GluA3[WT]. Data are normalized to the mean WT synapse volume per group for GluA2, GluA4, or GluA_Sum. (**H**) Data in panel G displayed as cumulative distributions. Instead of normalizing to the WT group mean as in panels **E–G**, here data were normalized to each image mean to visualize differences in the shape of the distributions between GluA3[WT] and GluA3[KO].

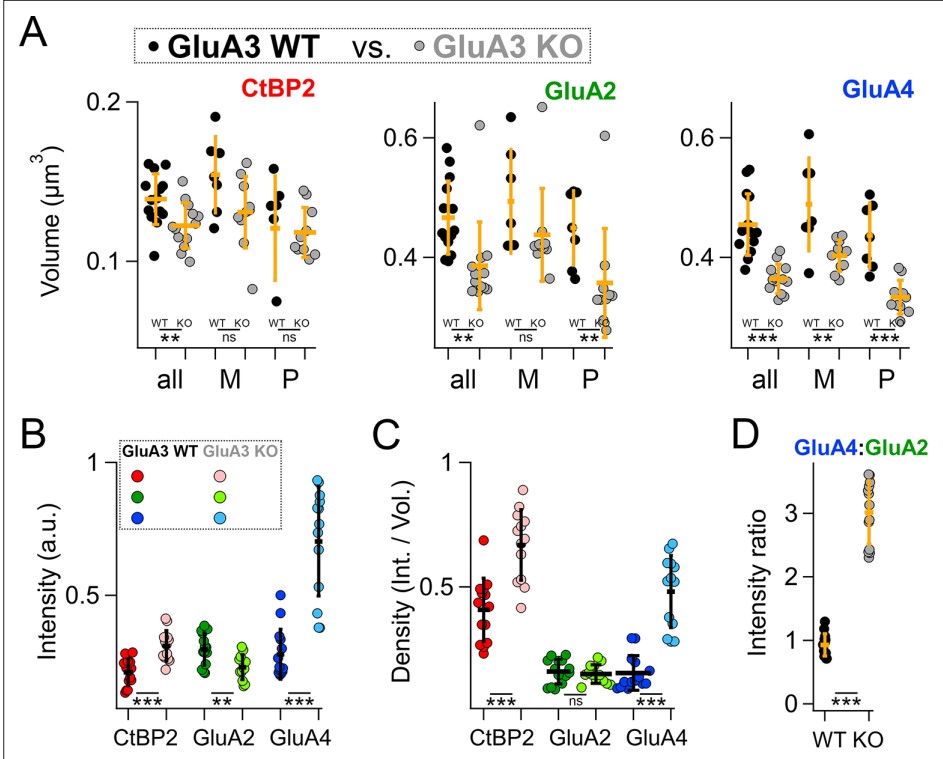

**Figure 8.** Modiolar- and pillar-side volume, intensity, and density of presynaptic ribbon and postsynaptic α-amino-3-hydroxy-5-methyl-4-isoxazole propionic acid receptor (AMPAR) subunits. (**A**) Quantification of CtBP2, GluA2, or GluA4 mean volume per image for GluA3$^{WT}$ (black, $n$ = 2990 synapses from 14 images) compared to GluA3$^{KO}$ (gray, $n$ = 2814 synapses from 14 images). Each point represents an image mean. Gold bars are mean ± standard deviation (SD). For CtBP2, there is an overall reduction in volume in GluA3$^{KO}$ ($p$ = 0.008, $U$: 144, $n_{WT}$ = 14, $n_{KO}$ = 13). For GluA2, the overall volume reduction in GluA3$^{KO}$ ($p$ = 0.0001, $U$: 168, $n_{WT}$ = 14, $n_{KO}$ = 13) resulted from smaller puncta on the pillar side of GluA3$^{KO}$ relative to GluA3$^{WT}$ ($p$ = 0.0083; $U$: 90, $n_{WT}$ = 7, $n_{KO}$ = 10) but not on the modiolar side ($p$ = 0.063). For GluA4, the overall volume reduction in GluA3$^{KO}$ ($p$ = 4.9e$^{-6}$; $U$: 176, $n_{WT}$ = 14, $n_{KO}$ = 13) resulted from smaller puncta on the pillar side of GluA3$^{KO}$ ($p$ = 0.0004; $U$: 96, $n_{WT}$ = 7, $n_{KO}$ = 10) and on the modiolar side of GluA3$^{KO}$ ($p$ = 0.0058; $U$: 62, $n_{WT}$ = 7, $n_{KO}$ = 10) relative to GluA3$^{WT}$. See ***Figure 8—figure supplement 1***. (**B**) Quantification of median intensities per image for data in panel A. CtBP2 intensity increased in GluA3$^{KO}$ ($p$ = 0.0001; $U$: 17, $n_{WT}$ = 14, $n_{KO}$ = 13); GluA2 intensity decreased in GluA3$^{KO}$ ($p$ = 0.01; $U$: 143, $n_{WT}$ = 14, $n_{KO}$ = 13); and GluA4 intensity decreased in GluA3$^{KO}$ ($p$ = 5e$^{-6}$; $U$: 6, $n_{WT}$ = 14, $n_{KO}$ = 13). (**C**) Increase in CtBP2 ($p$ = 5e$^{-5}$; $U$: 14, $n_{WT}$ = 14, $n_{KO}$ = 13) and GluA4 median density per synapse ($p$ = 5e$^{-6}$; $U$: 6, $n_{WT}$ = 14, $n_{KO}$ = 13) in GluA3$^{KO}$ relative to GluA3$^{WT}$, but not GluA2 ($p$ = 0.63; $U$: 101, $n_{WT}$ = 14, $n_{KO}$ = 13). (**D**) Increase in GluA4:GluA2 intensity ratio in GluA3$^{KO}$ relative to GluA3$^{WT}$ ($p$ = 6e$^{-7}$; $U$: 0, $n_{WT}$ = 14, $n_{KO}$ = 13). Mann-Whitney two-tailed U-test; ** $p$ < 0.01, *** $p$ < 0.001, ns: not significant.

The online version of this article includes the following source data and figure supplement(s) for figure 8:

**Source data 1.** Confocal data and statistics of synapse volumes and intensities for GluA3$^{WT}$ and GluA3$^{KO}$.

**Figure supplement 1.** Modiolar-, pillar-side groupings and example synapses from GluA3$^{WT}$ and GluA3$^{KO}$.

pillar-side synapses were significantly smaller in GluA3$^{KO}$ (M: 0.40 ± 0.026 μm$^3$; P: 0.33 ± 0.028 μm$^3$) relative to GluA3$^{WT}$ (M: 0.49 ± 0.078 μm$^3$, $p$ = 0.0058; P: 0.44 ± 0.055 μm$^3$, $p$ = 0.0004).

In contrast to mean CtBP2 volume, which was decreased in GluA3$^{KO}$ (***Figure 8A***), median CtBP2 intensity (***Figure 8B***), and density (***Figure 8C***) were significantly increased in GluA3$^{KO}$. For CtBP2 intensity (a.u.)e$^5$ – GluA3$^{WT}$: 1.2 ± 0.3; *GluA3$^{KO}$*: 1.8 ± 0.3, $p$ = 0.0001. As shown in the representative images assessed in ***Figure 7***, GluA2 intensity and volume were both reduced (***Figure 8A, B***), resulting in no change in GluA2 density (***Figure 8C***). For GluA2 intensity (a.u.)e$^5$ – GluA3$^{WT}$: 1.8 ± 0.4; GluA3$^{KO}$: 1.4 ± 0.3, $p$ = 0.01. While GluA4 intensity increased, volume was reduced (***Figure 8A, B***), resulting in increased GluA4 density (***Figure 8C***). For GluA4 intensity (a.u.)e$^5$ – GluA3$^{WT}$: 1.7 ± 0.6; GluA3$^{KO}$:

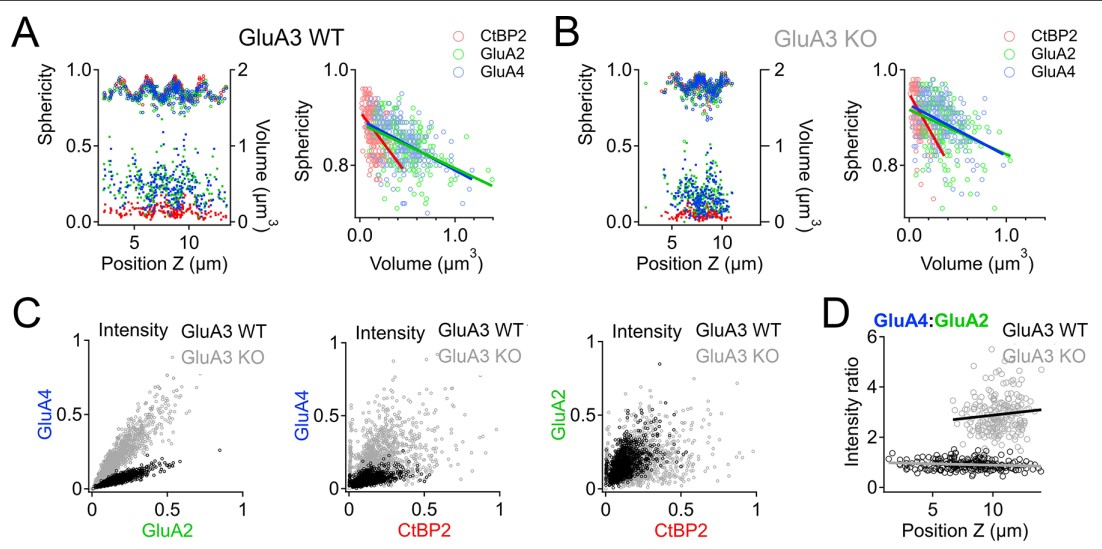

**Figure 9.** Spatial trends of synapse sphericity, volume, and α-amino-3-hydroxy-5-methyl-4-isoxazole propionic acid receptor (AMPAR) subunit relative abundance in the organ of Corti. (**A**) Volume and sphericity per synapse vs. Z-axis position for an exemplar GluA3[WT] image from *Figure 7* showing spatial oscillations in CtBP2, GluA2, and GluA4. Right: Inverse relationship between synapse sphericity and volume for CtBP2, GluA2, and GluA4. (**B**) For GluA3[KO], as in panel A. (**C**) Left: Normalized intensity of GluA4 vs. normalized intensity of GluA2 for GluA3[WT] (black) and GluA3[KO] (gray). Center: Normalized intensity of GluA4 vs. normalized intensity of CtBP2. Right: Normalized intensity of GluA2 vs. normalized intensity of CtBP2. (**D**) GluA4:GluA2 intensity ratio vs. Z-axis position. Panels C and D for six WT and six KO images from the mid-cochlea.

4.2 ± 1.2, p = 5e[−6]. Relative to GluA3[WT] synapses, the GluA4:GluA2 intensity ratio was 3× greater on average for GluA3[KO] synapses (*Figure 8D*). For GluA4:GluA2 intensity ratio – GluA3[WT]: 0.93 ± 0.18; GluA3[KO]: 3.0 ± 0.5, p = 6e[−7].

## Positive correlations between synaptic puncta volumes, intensities, and sphericities in GluA3[WT] are reduced in GluA3[KO] as the range of modiolar–pillar positions is shortened

In GluA3[WT], we commonly observed apparent oscillations in synapse volume as a function of position in the Z-axis of the confocal microscope when the modiolar–pillar dimension was approximately parallel to the Z-axis (*Figure 9A*, left panel, right axis, lower markers). These spatial oscillations were clearer when measured as sphericity (*Figure 9A*, left panel, left axis, upper markers), which was inversely related to volume (*Figure 9A*, right panel). We observed a similar phenomenon in GluA3[KO] (*Figure 9B*), although the synapses resided in a smaller range along the Z-axis. GluA2 and GluA4 intensities per synapse were positively related in both genotypes (*Figure 9C*, left). GluA2 and GluA4 intensities were positively related with CtBP2 intensities, but the relationships were less apparent in GluA3[KO] (*Figure 9C*, center and right), consistent with the increase in CV measured for GluA2 and GluA4 intensities per synapse in GluA3[KO] relative to GluA3[WT] (*Figure 7*). Plotting the GluA4:GluA2 intensity ratio as a function of Z-position revealed that increases of the GluA4:GluA2 intensity ratios in GluA3[KO] relative to GluA3[WT] tended to be greater for synapses on the pillar side than the modiolar side (*Figure 9D*). In summary, synapse alterations in GluA3[KO] were greatest on the pillar side, suggesting defective molecular differentiation of synapse properties along the morphological modiolar–pillar axis.

## Discussion

Hearing depends on the activation of AMPARs on the postsynaptic terminals of auditory nerve fibers (*Ruel et al., 1999*; *Glowatzki and Fuchs, 2002*). Cochlear AMPARs are tetrameric heteromers comprised of the pore-forming subunits GluA2, 3, and 4, where the absence of GluA2 results in a CP-AMPAR channel with increased permeability to $Ca^{2+}$ and $Na^+$. AMPAR tetramers assemble as dimers of dimers, with the GluA2/3 dimer being energetically favored and prominent in the brain

(*Greger and Mayer, 2019*). Our study shows that postsynaptic GluA3 subunits are required for the appropriate assembly of AMPAR GluA2 and GluA4 subunits on mammalian cochlear afferent synapses. Remarkably, we find that GluA3 is also essential for presynaptic ribbon modiolar–pillar morphological distinctions. We propose that postsynaptic GluA3 subunits at IHC-ribbon synapses may perform an organizational function beyond their traditional role as ionotropic glutamate receptors.

In the absence of GluA3 subunits, we observed a reduction in GluA2 subunits and an increase in GluA4 subunits at the ribbon synapses of 5-week-old male mice (*Figures 6–9*), preceding the decline of cochlear output observed at 8 weeks (*García-Hernández et al., 2017*). Taken together with previous work, we hypothesize that the 5-week-old GluA3$^{KO}$ cochlea of male mice is in a pathological but presymptomatic, vulnerable state. In GluA3$^{WT}$ mice on C57BL/6 background, synapse components (ribbons, PSDs, and SVs) tended to be larger on the modiolar side in electron microscopy (*Figures 3–5*). Without GluA3 the pillar-side synaptic components were enlarged in GluA3$^{KO}$ relative to GluA3$^{WT}$, including a ~50% increase in mean PSD surface area (*Figure 5A*), but the immunofluorescent volumes of synaptic AMPARs were concomitantly reduced on the pillar side in GluA3$^{KO}$ relative to the modiolar side and relative to GluA3$^{WT}$ (*Figure 8A*). This mismatch between changes in PSD size and changes in AMPAR cluster-size suggests misregulation of AMPAR density in the PSD in the absence of GluA3. In mice, SGN fibers contact IHCs in the differentiating organ of Corti already at birth. Synaptic ribbons and synaptic membrane densities mature over the following 3–4 weeks (*Sobkowicz et al., 1986*; *Wong et al., 2014*; *Michanski et al., 2019*; *Payne et al., 2021*). Future studies should determine if these changes in synapse ultrastrucure and molecular composition result from disrutption of embryonic or early postnatal development and/or later postnatal maturation.

## GluA3 is required for appropriate AMPAR assembly at IHC-ribbon synapses

Noise-induced cochlear synaptopathy is caused by excitotoxic over-activation of AMPARs due to excessive glutamate release from the sensory IHCs (*Puel et al., 1998*; *Kim et al., 2019*). Antagonizing the Ca$^{2+}$-permeable subset of AMPARs (CP-AMPARs) pharmacologically can prevent noise-induced synaptopathy while allowing hearing function to continue through activation of Ca$^{2+}$-impermeable AMPARs (*Hu et al., 2020*). In the absence of GluA3, GluA2/4 would be the only heterodimer. Homodimers can form and homomeric tetramers may exist, but non-GluA2 subunits preferentially heterodimerize with GluA2 subunits because homodimers are less stable energetically (*Rossmann et al., 2011*; *Zhao et al., 2017*), suggesting that GluA2/4 heterodimers should be predominant in GluA3$^{KO}$. However, we find that loss of GluA3 alters GluA2 and GluA4 subunit relative abundance, increasing the GluA4:GluA2 ratio (*Figures 8D and 9D*), which may increase the number of GluA2-lacking CP-AMPARs at cochlear ribbon synapses of the GluA3$^{KO}$ mice by forcing GluA4 homodimerization. The increase in CP-AMPARs in the GluA3$^{KO}$ could make the IHC-ribbon synapses more vulnerable to excitotoxic noise trauma as the cochlea matures and ages. In support of this, male GluA3$^{KO}$ mice have reduced ABR wave-1 amplitude relative to GluA3$^{WT}$ mice by 2 months of age and elevated ABR thresholds by 3 months of age (*García-Hernández et al., 2017*). Although young male GluA3$^{KO}$ mice have ABR and synapse numbers similar to WT (*Figures 1 and 6*), we hypothesize these molecular-anatomical alterations to AMPAR subunits result in synapses with increased vulnerability to AMPAR-mediated excitotoxicity that lead to synapse loss and hearing loss as the mice age in ambient sound conditions.

It is important for future studies to determine the effects on cochlear ribbon synapses when GluA3 is removed after cochlear maturation, in comparison with the present study in which GluA3 is absent congenitally resulting in direct and indirect effects. How are the amplitude and kinetics of synaptic transmission altered under these conditions? If two synapses have the same number of AMPARs, the synapse with a greater proportion of CP-AMPARs would have increased postsynaptic current amplitude. However, the increase in GluA4 seems to be outweighed by the reduction in GluA2 and the absence of GluA3 when measuring total AMPAR volume or fluorescence per synapse, suggesting fewer AMPARs per synapse in GluA3$^{KO}$ (*Figure 8*). In the context of homeostatic synaptic plasticity (*Turrigiano, 2012*), an increase in synaptic current size due to increased proportion of CP-AMPARs could be compensated by net removal of AMPARs. Thus, future studies will need to determine if these synapses suffer from postsynaptic currents that are too large, too small, or similar in size with altered single-channel conductance.

Most knowledge of AMPAR subunit composition comes from studies on neurons in the brain, where GluA1 subunits are highly expressed. For example, AMPARs of hippocampal CA1 neuronal synapses are predominantly comprised of GluA1/2 heteromers (~80%; *Lu et al., 2009*). Cochlear AMPARs are unique because mature SGNs do not express GluA1 (*Niedzielski and Wenthold, 1995*; *Matsubara et al., 1996*; *Parks, 2000*; *Shrestha et al., 2018*; this study). In the ascending auditory pathway, the presynaptic terminals of SGNs release glutamate onto neurons in the brainstem's cochlear nucleus where, as in the cochlea, AMPARs appear to lack GluA1 (*Figure 1B*, right). In the cochlea and cochlear nucleus, elimination of GluA3 did not result in GluA1 expression (*Figure 1B*, left and right). Given the influences of GluA1 subunits in AMPAR activity-dependent plasticity in the brain (*Lee et al., 2010*), and considering their absence in the mature cochlea, the rules governing AMPAR dynamics in the inner ear are largely unknown and likely to be unique.

The duration of the postsynaptic current and thus the net influx of charge is largely determined by AMPAR desensitization, which is regulated by the proportion of *flip* and *flop* splice variants of GluA subunits (*Sommer et al., 1990*; *Monyer et al., 1991*; *Mosbacher et al., 1994*; *Koike et al., 2000*). *Flip* subunits desensitize more slowly than *flop* subunits, resulting in slower decays of AMPAR-mediated currents (*Trussell and Fischbach, 1989*; *Lawrence and Trussell, 2000*; *Gardner et al., 2001*; *Quirk et al., 2004*; *Pei et al., 2007*). Changes in transcription and mRNA splicing of AMPAR subunits may influence excitotoxicity. For example, a decrease in GluA2 and GluA3 *flop* isoforms leads to elevated intracellular $Ca^{2+}$ levels and increased death of retina ganglion cells after glucose deprivation (*Park et al., 2016*). At synapses between auditory nerve fibers (endbulbs) and bushy cells in the cochlear nucleus, the lack of GluA3 results in a lower proportion of *flop* subunits (*Antunes et al., 2020*). In contrast, we find the absence of GluA3 did not significantly alter transcription and mRNA splicing of GluA2 and GluA4 isoforms in the cochleae of 5-week-old GluA3[KO] mice, arguing against the possibility that such alterations underlie the pathology of IHC-ribbon synapses in GluA3[KO] mice (*Figure 1*). We note that our method of isoform analysis was from whole temporal bones, including outer hair cell synapses and vestibular hair cell synapses. Contamination from outer hair cell synapses is likely to be small due to low AMPAR expression levels (*Weisz et al., 2012*; *Martinez-Monedero et al., 2016*). However, the presence of the vestibular synapses and/or the mixing of synapses from the whole cochlea may have prevented us from detecting changes in isoform expression at the level of individual synapses. Overall, in the inner ear in the absence of GluA3, *flop* isoforms of GluA2 and GluA4 were more abundant than *flip* isoforms, as in WT. Future patch-clamp studies are required to test AMPAR desensitization in the absence of GluA3 at the cellular level.

## Potential trans-synaptic role of GluA3 at IHC-ribbon synapses

Pre- and postsynaptic ultrastructural features of IHC-ribbon synapses are disrupted in the organ of Corti of GluA3[KO] mice (*Figures 3–5*). This is reminiscent of the ultrastructure of endbulb synapses in the cochlear nucleus of GluA3[KO] mice, which is altered due to trans-synaptic developmental effects, where the absence of GluA3 increases synaptic depression by increasing the initial probability of vesicle release, slowing vesicle replenishment, and reducing the readily releasable pool of SVs through unknown mechanisms (*García-Hernández et al., 2017*; *Antunes et al., 2020*). In the cochlea, modiolar–pillar ultrastructural differences among ribbon synapses were eliminated or reversed in GluA3[KO]. Our ultrastructural analysis shows the absence of GluA3 resulted in loss of the modiolar–pillar difference in PSD surface area seen in GluA3[WT], due to larger PSDs on the pillar side of GluA3[KO] relative to GluA3[WT]. At endbuld synapses, an increase in PSD surface area or thickness, and changes in SVs size have been observed in congenitally deaf cats and mice (*Lee et al., 2003*; *Ryugo et al., 1997*; *Ryugo et al., 2005*), and after conductive hearing loss (*Clarkson et al., 2016*). Similarly, the increase in PSD surface area and larger SVs of pillar-side synapses may represent initial pathological structural signs in the absence of GluA3.

Through development, the ribbon-shape changes from largely round to oval, droplet-like, or wedge-like shapes (*Wong et al., 2014*; *Michanski et al., 2019*). The absence of GluA3 during development resulted in ribbons with shorter long axes that were more spherical relative to GluA3[WT]. This finding is consistent with a developmental defect in a process of ribbon maturation, whereby modiolar-side ribbons become longer and less spherical between 2.5 and 5 weeks of age in C57BL/6 WT mice (*Payne et al., 2021*). While GluA3[KO] ribbons were shorter in long axis and more rounded in transmission electron microscopy (TEM) than those from GluA3[WT], they also tended to have larger

volumes particularly on the pillar side, suggesting lengthening of the short ribbon axis in GluA3$^{KO}$. Finally, loss of GluA3 eliminated the modiolar–pillar difference in SV size observed in WT, due to an increase in SV size on the pillar side of GluA3$^{KO}$. We note that other studies of IHC-ribbon synapses have identified increased SV size as a phenotype in endophilin KO and in AP180 KO mice (*Kroll et al., 2019*; *Kroll et al., 2020*), although modiolar and pillar comparisons were not made.

Our ultrastructural data suggest that postsynaptic GluA3 subunits at IHC-ribbon synapses may perform an organizational function beyond their traditional role as ionotropic glutamate receptors. The mechanisms are still unclear, but evidence shows that AMPARs convey a retrograde trans-synaptic signal essential for presynaptic maturation (*Tracy et al., 2011*). AMPAR subunits in the cochlea may interact with the trans-synaptic adhesion factors Neuroligins and Neurexins (*Heine et al., 2008*; *Hickox et al., 2017*; *Ramirez et al., 2022*). GluA3 is required for the functional development of the presynaptic terminal and the structural maturation of SV size in endbulb auditory nerve synapses in the cochlear nucleus (*Antunes et al., 2020*). Altered SV size together with a change in the number of AMPARs and their clustering at the synapse contribute to quantal size variation and altered synaptic transmission (*Levy et al., 2015*). The number of AMPARs at IHC-ribbon synapses is undetermined but with a synaptic surface area ranging 0.1–1.5 μm$^2$ (*Liberman, 1980*; *Payne et al., 2021*) or 0.3–0.8 μm$^2$ (current study), it is estimated several hundred to a few thousand AMPARs at each PSD (*Momiyama et al., 2003*). In GluA3$^{KO}$ mice, we find that despite the decrease in GluA2 and the larger increase of GluA4, the overall intensity and volume of the AMPAR subunit immunolabeling at IHC-ribbon synapses decreases when compared to WT synapses, primarily due to loss of GluA3. An increase in relative abundance of CP-AMPARs and a decreased overall abundance of AMPARs in GluA3$^{KO}$ are expected to have opposing effects on the size of the synaptic current evoked by glutamate. The ABR wave-1 amplitude is unaltered in male GluA3$^{KO}$ at 5 weeks of age suggesting a similar hearing sensitivity to WT mice. However, ABR peak amplitudes are reduced in the male KO at 8 weeks of age (*García-Hernández et al., 2017*). To strengthen and confirm the potential trans-synaptic role of GluA3 at IHC-ribbon synapses and to compare synaptic strength, further electrophysiological studies need to determine the existence of altered quantal size and quantal content in the GluA3$^{KO}$.

In the cochlea, afferent synaptic contact formation on the IHC, characterized by a demarcated pre- and postsynaptic density, often precedes ribbon attachment at the presynaptic active zone (AZ) membrane. Ribbon attachment occurs around embryonic day 18 (E18) (*Michanski et al., 2019*). The presence of postsynaptic AMPARs in those embryonic IHC-ribbon synapses has not been reported. However, patches of GluA2/3 AMPAR subunit immunolabeling were observed during the first postnatal week, juxtaposed to the presynaptic ribbon marker RIBEYE (*Wong et al., 2014*). Fusion of ribbon precursors extends after hearing onset and is a critical step in presynaptic AZ formation and maturation (*Michanski et al., 2019*). This maturation of synaptic ribbons may be essential for the functional maturation of afferent synaptic transmission within the cochlea. In mice lacking synaptic ribbons, features like PSDs, presynaptic densities, voltage-gated Ca$^{2+}$ channels, and bassoon seem to develop relatively independently of ribbon presence, but spike patterns in the auditory nerve fibers are altered (*Jean et al., 2018*; *Becker et al., 2018*). It is possible that GluA3 plays a direct or indirect role in the recruitment and maintenance of pre- and postsynaptic proteins, for example, via its N- and C-terminus domains. Postsynaptic PDZ domain AMPAR C-terminus interacting proteins such as PSD95 are present at IHC-ribbon synapses early during postnatal development (*Tong et al., 2013*; *Wong et al., 2014*). PSD95 interacts with the cell adhesion proteins Neuroligins (*Irie et al., 1997*; *Jeong et al., 2019*). Neuroligin-3 and to a lesser extent Neuroligin-1 are essential to cochlear function (*Ramirez et al., 2022*). In the CNS, alignment of postsynaptic AMPARs, PSD95, and Neuroligin-1 together with the presynaptic protein RIM form nanocolumns (*Tang et al., 2016*) thought to represent essential elements of trans-synaptic connections. The existence of such nanocolumns at ribbon synapses, where several release sites may reside around each ribbon, is unknown, but the same proteins are present (*Jung et al., 2015*; *Krinner et al., 2017*; *Picher et al., 2017*; *Hickox et al., 2017*; *Ramirez et al., 2022*). To understand further the synaptic mechanisms of hearing and hearing loss, it will be essential to identify the full complement of cleft-spanning adhesion proteins that interact with AMPARs at IHC synapses, in particular with GluA3.

# Materials and methods

## Animals

A total of 26 C57BL/6 wild-type (GluA3$^{WT}$, $n$ = 13) and GluA3-knockout (GluA3$^{KO}$, $n$ = 13) mice were used in this study. The *Gria3* gene encodes the GluA3 protein, one of four AMPAR pore-forming subunits (GluA1–4) encoded by four genes *Gria1–4*. Generation of the GluA3$^{KO}$ mice has been previously described (*García-Hernández et al., 2017*; *Rubio et al., 2017*). Male WT and KO mice from two separate colonies were compared at 5 weeks of age following normal rearing in an animal facility with 55–75 dB SPL ambient sound pressure level over time (unweighted mean SPL, 2–80 kHz, Sensory Sentinel, Turner Scientific). Mice were fed ad libitum and maintained on a 12-hr light/12-hr dark daily photoperiod. All experimental procedures were in accordance with the National Institute of Health guidelines and approved by the University of Pittsburgh Institutional Animal Care and Use Committee.

## Auditory brainstem response (ABR)

To test the auditory output of the GluA3$^{WT}$ and GluA3$^{KO}$ mice, we performed ABR as previously described (*Clarkson et al., 2016*; *García-Hernández et al., 2017*; *García-Hernández and Rubio, 2022*; *Weisz et al., 2021*). Recordings were conducted under isoflurane anesthesia in a soundproof chamber and using a Tucker-Davis Technologies (Alachua, FL) recording system. Click or tone stimuli were presented through a calibrated multifield magnetic speaker connected to a 2-mm diameter plastic tube inserted into the ear canal. ABR were recorded by placing subdermal needle electrodes at the scalp's vertex, at the right pinna's ventral border, and the ventral edge of the left pinna. ABR were recorded in response to broadband noise clicks (0.1 ms) or tone pips of 4, 8, 12, 16, 24, and 32 kHz (5 ms). Stimuli were presented with alternating polarity at a rate of 21 Hz, with an interstimulus interval of 47.6 ms. The intensity levels used were from 80 to 10 dB, in decreasing steps of 5 dB. The waveforms of 512 presentations were averaged, amplified 20×, and digitalized through a low-impedance preamplifier. The digitalized signals were transferred via optical port to an RZ6 processor, where the signals were band-pass filtered (0.3–3 kHz) and converted to analog form. The analog signals were digitized at a sample rate of ~200 kHz and stored for offline analyses. Hearing threshold levels were determined from the averaged waveforms by identifying the lowest intensity level at which clear, reproducible peaks were visible. Wave-1 amplitudes were compared between GluA3$^{WT}$ and GluA3$^{KO}$ mice. For measurements of amplitudes, the peaks and troughs from the click-evoked ABR waveforms were selected manually in BioSigRZ software and exported as CSV files. The peak amplitude was calculated as the height from the maximum positive peak to the next negative trough.

## Immunohistochemistry and immunofluorescence

A total of 14 mice (GluA3$^{WT}$ $n$ = 7; GluA3$^{KO}$ $n$ = 7) were anesthetized with a mixture of ketamine (60 mg/kg) and xylazine (6.5 mg/kg) and were transcardially perfused with 4% paraformaldehyde (PFA) in 0.1 M phosphate buffer (PB) pH = 7.2. After 10 min of transcardial perfusion, cochleae and brains were removed from the skull and postfixed for 45 min on ice. At the beginning of postfixation the stapes was removed, a hole was opened at the apex of the cochlea bone shell, and each cochlea was perfused with the same fixative through the oval window. After postfixation, the cochleae and brains were washed in 0.1 M phosphate-buffered saline (PBS).

Four cochleae (2 of each genotype) were decalcified in 10% ethylenediaminetetraacetic acid (EDTA) in PBS for 24 hr, cryoprotected in 10%, 20%, and 30% sucrose in 0.1 M PBS, frozen on dry ice with tissue freezing medium (Electron Microscopy Sciences, Hatfield, PA), and stored at −20°C for up to 1 month. Brains were cryoprotected in the same sucrose dilution gradient and frozen on dry ice. Cochleae were sectioned at 20-µm thickness with a cryostat and were mounted on glass slides. Brains were cut with a slicing vibratome at 50–60 µm thickness and collected on culture wheel plates containing 0.1 M PBS. Cochlea and brain sections followed standard immunofluorescence and immunohistochemistry protocols described in *Douyard et al., 2007* and *Wang et al., 2011*. Primary rabbit polyclonal antibodies against GluA2 (Millipore, AB1768; RRID:AB_2247874), GluA4 (Millipore, AB1508; RRID:AB_90711), and GluA1 (a gift from Robert J. Wenthold; *Douyard et al., 2007*) were used at a 1:500 dilution in 0.1 M PBS. Cochlear sections were incubated with an Alexa-594 goat-anti-rabbit secondary antibody (1:1000; Life Tech.). Brain slices were incubated in a biotinylated secondary antibody goat anti-rabbit (1:1000; Jackson Laboratories) in 0.1 M PBS. Then, brain sections were incubated in avidin–biotin–peroxidase complex (ABC Elite; Vector Laboratories; 60 min; room

temperature [RT]), washed in 0.1 M PBS, and developed with 3,3-diaminobenzidine plus nickel (DAB; Vector Laboratories Kit; 2–5 min reaction). Sections were analyzed with an Olympus BX51 upright microscope, and digital images were captured with the CellSens software (Olympus S.L.).

The other ten cochleae (five of each genotype) were shipped overnight to Washington University in Saint Louis in 0.1 M PBS containing 5% glycerol for wholemount immunolabeling and confocal analysis of presynaptic ribbons (CtBP2/Ribeye) and postsynaptic AMPAR subunits GluA2, GluA3, and GluA4 as previously described (*Jing et al., 2013*; *Ohn et al., 2016*; *Sebe et al., 2017*; *Kim et al., 2019*; *Hu et al., 2020*). Primary antibodies: CtBP2 mouse IgG1 (BD Biosciences 612044; RRID:AB_399431), GluA2 mouse IgG2a (Millipore MAB397; RRID:AB_2113875), GluA3 goat (Santa Cruz Biotechnology SC7612), and GluA4 rabbit (Millipore AB1508; RRID:AB_90711) were used with species-appropriate secondary antibodies conjugated to Alexa Fluor (Life Tech.) fluorophores excited by 488, 555, or 647 nm light in triple-labeled samples mounted in Mowiol. Samples were batch processed using the same reagent solutions in two cohorts, each including WT and KO mice. Although Southern blot, western blot, and freeze-fracture postembedding immunogold labeling confirmed a lack of *Gria3* DNA or GluA3 protein subunit expression in GluA3$^{KO}$ mice (*García-Hernández et al., 2017*; *Rubio et al., 2017*; in those two studies an antibody against the N-terminus domain of GluA3 was used), we found that the polyclonal GluA3 C-terminal antibody (SC7612) showed weak binding to cochlear synapses in GluA3$^{KO}$ tissue. This false 'anti-GluA3' signal on synapses in the KO presumably results from binding of the GluA3 antibody to the GluA2 C-terminus, as stated in the Santa Cruz specification sheet (https://datasheets.scbt.com/sc-7612.pdf), due to sequence similarity between GluA2 and GluA3 (*Dong et al., 1997*). Although this artifact in GluA3$^{KO}$ is not visible when the brightness levels are set to avoid saturation in the corresponding GluA3$^{WT}$ images, we measured the false signal in GluA3$^{KO}$ and estimated it to be ~10% of the signal in GluA3$^{WT}$ (not shown). Thus, approximately 10% of the signal quantified as anti-GluA3 immunofluorescence in GluA3$^{WT}$ may be cross-reactivity with GluA2.

## Confocal microscopy and image analysis

For synapse counts and measurements of intensity, volume, sphericity, and position confocal stacks were acquired with a *Z*-step of 0.37 µm and pixel size of 50 nm in *X* and *Y* on a Zeiss LSM 700 with a 63 × 1.4 NA oil objective lens. To avoid saturation of pixel intensity and to enable comparisons across images and genotypes, we first surveyed the samples to determine the necessary laser power and gain settings to collect all of the images. Then, using identical acquisition settings for each sample, we collected three to four images from each cochlea at each of the three cochlear regions (basal, middle, and apical, respectively) centered near tonotopic characteristic frequencies of 40, 20, and 10 kHz (*Müller et al., 2005*). For display only, the brightness and contrast levels were adjusted linearly in *Figures 2 and 6* for visual clarity. Image analysis was performed on unadjusted, raw data.

The numbers of hair cells and paired and unpaired pre- and postsynaptic puncta were counted and manually verified after automated identification using Imaris software (Bitplane) to calculate the mean per IHC per image. The observers were blinded to mouse genotype. For each group of images from which synapses were counted or synaptic properties were measured, grand means ( ± SD) were calculated across image means (*Figures 6B, 7E, G* and, *Figure 8A–D*). Paired synapses were identified as juxtaposed puncta of presynaptic ribbons (CtBP2) and postsynaptic AMPARs (GluA2 and/or GluA4), which appear to partly overlap at confocal resolution (*Rutherford, 2015*). Unpaired (i.e., lone) ribbons were defined as CtBP2 puncta in the IHC but lacking appositional GluA2 or GluA4 puncta. For unpaired ribbons, we did not distinguish membrane anchored from unanchored. Ribbonless synapses consisted of GluA2 and/or GluA4 puncta located around IHC basolateral membranes but lacking CtBP2. Pixels comprising puncta of synaptic fluorescence were segmented in 3D as 'surface' objects in Imaris using identical settings for each image stack, including the 'local contrast background subtraction' algorithm for automatically calculating the threshold pixel intensity for each fluorescence channel in each image. This adaptive and automatically calculated thresholding algorithm compensated for differences in overall luminance between image stacks that would affect the volume of segmented puncta if a fixed threshold was applied across images, and avoided the potential subjective bias of setting a user-defined arbitrary threshold value separately for each image. Intensity per synaptic punctum was calculated as the summation of pixel intensities within the surface object. To associate the intensities of different GluA puncta belonging to the same synapse (*Figures 7C and*

9C-D), we generated surface objects from a virtual fourth channel equal to the sum of the three channels (GluA2, 3, and 4; or CtBP2, GluA2, and GluA4) and then summated the pixel intensities within each of the three fluorescence channels comprising each synapse segmented as a punctum on the fourth channel. The average density of synaptic fluorescence per image (*Figure 8C*) was calculated as median punctum Intensity (a.u.) divided by median punctum Volume ($\mu m^3$) using surface objects calculated from corresponding individual fluorescence channels. To associate the volumes of different GluA puncta belonging to the same synapse (*Figure 7B*), we used the virtual fourth channel to generate masks. The mask for each synapse had a unique color value. Objects belonging to the same synapse were identified based on common overlap with the unique color value assigned to each mask. Sphericity is the ratio of the surface area of a sphere to the surface area of an object of equal volume, so sphericity of 1 means the object is a perfect sphere. To differentiate synapse position, images were used in which the row of IHCs was oriented with the modiolar–pillar dimension in the microscope's Z-axis and the organ of Corti was not sloping in the image volume, and modiolar- and pillar-side groups were split at the midpoint of the range of synapses along the modiolar–pillar dimension.

## Reverse transcription-polymerase chain reaction and quantitative PCR

Under isofluorane anesthesia, mice (GluA3$^{WT}$ $n = 4$; GluA3$^{KO}$ $n = 4$) were euthanized via cervical dislocation and decapitation. Immediately after the following decapitation, the cranium was opened, and the inner ears were removed. Inner ears were flash-frozen in liquid nitrogen and stored in −80°C for up to 1 month until reverse transcription-polymerase chain reaction (RT-PCR). In preparation for RT-PCR, both inner ears from each individual were homogenized by hand with mortar or pestle and RNA was extracted with Trizol (Ambion by Life Technology). The RNA pellet ($n = 3$; each pellet contained the two cochleae of one mouse, except one contained the cochleae of two mice) was resuspended and the supernatant containing RNA from each individual's inner ears was prepared for RT-PCR using the SuperScript Strand Synthesis System kit (Invitrogen, cat. No. 11904018). The resulting cDNA was stored at −20°C for 1 week or less before real-time quantitative PCR (qPCR). qPCR was performed at the Genomics Research Core at the University of Pittsburgh using EvaGreen qPCR kit (MidSci, Valley Park, MO, cat. No. BEQPCR_R) and primers for *Gria2* and *Gria4 flip* and *flop* (*Gria2* and *Gria4* encode the GluA2 and GluA4 protein, respectively), which were the same primers used successfully in a previous RT-qPCR experiment by *Hagino et al., 2004*. In a 25-μl PCR reaction mixture, 2 μl cDNA samples were amplified in a Chromo 4 detector (MJ Research, Waltham, MA). GAPDH or 18S rRNA were used as housekeeping genes. Each sample (consisting of RNA product of both cochleae from each mouse) was run in triplicate, and average cycle thresholds (CTs) were used for quantification. Relative abundances of each splice isoform for GluA3$^{KO}$ males compared to GluA3$^{WT}$ were reported as fold change, calculated using the following equation: 2Delta-CT ($2^{-\Delta\Delta CT}$), where $\Delta\Delta CT = (CT_{GluA3}{}^{WT} - CT_{GAPDH}$ or $CT_{18S\ rRNA}) - (CT_{GluA3}{}^{KO} - CT_{GAPDH}$ or $CT_{18S\ rRNA})$, and CT represents the cycle threshold of each cDNA sample. For a more in-depth explanation of this equation see *Schmittgen and Livak, 2008*. Electrophoresis of 10 μl of RT-PCR products was performed using 3% agarose (SeaKem LE Agarose by Lonza) with molecular ladder gel containing 0.5 μg/ml ethidium bromide in 0.5× tris-acetate-ethylenediaminetetraacetic acid (TAE) buffer (pH 8.0) and run at 100 V for 60 min. The size and thickness of the agarose gel, reagents, and other conditions were kept constant. The band size and DNA concentration of each PCR amplicon were determined by comparison to the corresponding band in the molecular weight ladder (Gene Ruler 100 BP DNA ladder Thermo Scientific). The amplicon images (RT-PCR bands) in the gel were captured under ultraviolet light and documented using a Bio Rad Molecular Imager Gel Doc RX + Imaging system. All the parameters and experimental conditions used were kept constant throughout the study. The image was saved (in JPEG format) on a computer for digital image analysis using ImageJ software. The mean gray value (MGV) of each band was determined with NIH-ImageJ software (https://imagej.nih.gov/ij/). Samples were normalized to GAPDH. The *flip/flop* ratio was obtained by dividing the MGVs of the *flip* by the *flop*.

## Transmission electron microscopy

Four mice (2 per genotype) were anesthetized with a mixture of ketamine (60 mg/kg) and xylazine (6.5 mg/kg) and were transcardially perfused with 0.1 M PB, followed by 3% PFA and 1.5% glutaraldehyde in 0.1 M PB. Cochleae were dissected from the temporal bones, and fixative was slowly introduced through the oval window after removing the stapes and opening a hole at the apex of

the cochlea bone shell. Cochleae were postfixed overnight in the same fixative at 4°C and followed a protocol slightly modified from *Clarkson et al., 2016*. After decalcification in 10% EDTA for 24 hr at 4°C on a rotor, cochleae were washed in 0.1 M cacodylate buffer and postfixed with 1% osmium and 1.5% potassium ferrocyanide in cacodylate buffer for 1 hr at RT. Cochleae were dehydrated in an ascending ethanol gradient (ETOH; 35%, 50–70%, 80–90%) and were blocked stained with 3% uranyl acetate in 70% ETOH for 2 hr at 4°C before the 80% ETOH. The last dehydration steps performed with 100% ETOH and propylene oxide were followed by infiltration with epoxy resin (EMBed-812; Electron Microscopy Science, PA, USA). Cochleae were cut with a Leica EM UC7 ultramicrotome, and series of 15–20 serial ultrathin sections (70–80 nm in thickness) were collected. Each serial ultrathin section was collected on numbered single slot gold-gilded grids with formvar. Ultrathin sections were observed with a JEOL-1400 transmission electron microscope (TEM; JEOL Ltd., Akishima Tokyo, Japan), and images (at ×40,000 magnification) of the mid-cochlea (~20 kHz) containing IHC-ribbon synapses of the modiolar and pillar side were captured with an Orius SC200 CCD camera (Gatan Inc, Warrendale, PA, USA). In our ultrastructural analysis, we included IHC-ribbon synapses that were clearly located on either the modiolar face or the pillar face of the IHC. We did not sample every synapse in a given IHC, so the proportions of modiolar- and pillar-side synapses analyzed do not reflect the relative abundance of the population. The experimenter was not blinded to mouse genotype during image acquisition and analysis.

## 3D reconstructions and NIH ImageJ analysis of TEM micrographs

TEM micrographs (at ×40,000 magnification) of the serial IHC-ribbon synapses were calibrated, aligned, and reconstructed using Reconstruct software as previously described (*Gómez-Nieto and Rubio, 2009*; *Clarkson et al., 2016*; *Clarkson et al., 2020*; *https://synapseweb.clm.utexas.edu/software-0*; *Fiala, 2005*). A total of 29 (GluA3$^{WT}$) and 26 (GluA3$^{KO}$) IHC-ribbon synapses were reconstructed. In the 3D reconstructions, we used only the ultrathin sections of IHC-ribbon synapses containing the PSD. Sections containing the afferent dendrite without PSD were not included. In brief, two successive sections were aligned via rotation and translation such that corresponding structures like mitochondria in the two sections were superimposed. Linear transformation compensated for distortions introduced by the sectioning. Following alignment of the TEM sections, structures of interest were segmented visually into contours of separate objects. The thicknesses of all ultrathin sections in series were summed to account for differences in section thickness for 3D reconstructions. The subsequent linear interpolation between membrane contours in adjacent images resulted in polygonal outlines of cell membranes, PSD and synaptic ribbons. The 3D rendering was generated as VRML files from the stacks of all contoured sections. We calculated volumes and surface areas of the structures of interest (PSDs and ribbons) by filling these contours with tetrahedra. In addition, we used the NIH ImageJ software to determine the linear length of the PSD, the synaptic ribbon major axis, and SV size. For this analysis we used single representative TEM micrographs (at ×40,000 magnification) of each serial section series, as well as representative micrographs of other IHC-ribbon synapses that were not reconstructed because the serial section series were incomplete. A total of 43 GluA3$^{WT}$ and 53 GluA3$^{KO}$ PSDs were used to measure the linear length; 57 GluA3$^{WT}$ and 60 GluA3$^{KO}$ ribbons were used to calculate the major axes and circularity. For the SVs, we drew two perpendicular lines from the edges of the external membrane, to measure the major and minor diameters. The size of each SV was calculated as (Major diameter + Minor diameter)/2. Four to six SVs were analyzed per synapse. In *Figures 3–5*, we plotted the mean SV size per synapse (GluA3$^{WT}$ $n = 45$; GluA3$^{KO}$ $n = 47$) and compared grand means for modiolar and pillar groups.

## Statistical analysis

Statistical analyses were performed with GraphPad Prism software (Version 9.3.1) or IGOR Pro software (Wavemetrics, Version 7.08). Complete statistical details are included in source data files online for *Figures 1, 3–6* and *Figure 8*. One- or two-way ANOVA were used for comparisons of one or two independent variables, respectively. Two-tailed Mann–Whitney *U*-test and paired *t*-test were used to compare two independent groups, as indicated in the text. Paired and multiple comparisons were made using Šidák's and Tukey's tests, respectively, as indicated in source data files. Statistical significance for all tests was set to $p < \alpha$; $\alpha = 0.05$. Data are represented as mean ± standard deviation. The coefficient of variation (CV) was calculated as CV = SD/mean.

## Acknowledgements

This work was supported by NIDCD DC013048 (MER) and NIDCD DC14712 (MAR). We thank Nicholas Lozier for his helpful comments on the manuscript.

## Additional information

### Funding

| Funder | Grant reference number | Author |
|---|---|---|
| National Institutes of Health | DC013048 | Maria Eulalia Rubio |
| National Institutes of Health | DC14712 | Mark A Rutherford |

The funders had no role in study design, data collection, and interpretation, or the decision to submit the work for publication.

### Author contributions

Mark A Rutherford, Resources, Formal analysis, Supervision, Funding acquisition, Investigation, Methodology, Writing – review and editing, Data curation, Validation, Writing – original draft; Atri Bhattacharyya, Investigation, Visualization, Writing – review and editing; Maolei Xiao, Hou-Ming Cai, Investigation, Methodology; Indra Pal, Formal analysis, Investigation, Writing – review and editing; Maria Eulalia Rubio, Conceptualization, Resources, Data curation, Formal analysis, Supervision, Funding acquisition, Validation, Investigation, Visualization, Methodology, Writing – original draft, Project administration, Writing – review and editing

### Author ORCIDs

Mark A Rutherford ⓘ http://orcid.org/0000-0002-2627-6254
Maria Eulalia Rubio ⓘ http://orcid.org/0000-0003-3536-6013

### Ethics

This study was performed in strict accordance with the recommendations in the Guide for the Care and Use of Laboratory Animals of the National Institutes of Health. All of the animals were handled according to approved Institutional Animal Care and Use Committee (IACUC) protocols (#21100176 and #22030822) of the University of Pittsburgh. The protocol was approved by the Committee on the Ethics of Animal Experiments of the University of Pittsburgh (Permit Number: D16-00118). All auditory brainstem responses were measured under isoflurane anesthesia. All the transcardial perfusions were performed under ketamine and xylazine anesthesia, and every effort was made to minimize suffering.

### Decision letter and Author response

Decision letter https://doi.org/10.7554/eLife.80950.sa1
Author response https://doi.org/10.7554/eLife.80950.sa2

## Additional files

### Supplementary files

• MDAR checklist

### Data availability

All data analyzed during this study are included in the manuscript.

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
