## [Editor Report]

Hearing is mediated by hair cells in the cochlea, which synapse onto the primary dendrites of the auditory nerve. This study shows how deletion of a postsynaptic glutamate receptor subtype strongly influences inner hair cell-spiral ganglion cell synapse formation. This work shows that pre- and post-synaptic changes intertwine dynamically, providing insights into how pathological outcomes arise from synaptic perturbations.

---

## [Decision Letter]

**Decision letter after peer review:**

Thank you for submitting your article "Postsynaptic GluA3 subunits are required for the appropriate assembly of AMPA receptor GluA2 and GluA4 subunits on mammalian cochlear afferent synapses and for presynaptic ribbon modiolar-pillar morphological distinctions" for consideration by *eLife*. Your article has been reviewed by 2 peer reviewers, and the evaluation has been overseen by a Reviewing Editor and Barbara Shinn-Cunningham as the Senior Editor. The following individual involved in the review of your submission has agreed to reveal their identity: Chad Paul Grabner (Reviewer #1).

Essential revisions:

1. We suggest a supplemental table summarizing the data and the statistical tests used for each of the analyses

2. There is more background signal in wild type than ko ihcs in Figure 2A "sum". Is there an explanation for this? Please also address the GluA3 staining in the GluA3 KO (Figure 2C).

3. Revise the text to emphasize your IHC data, rather than the brainstem. Reviewers would like to read a discussion of your findings in the context of other studies of morphological/functional changes of synapses upon deletion AMPAR subunits (e.g. PMID: 19409270; PMID: 23664612; PMID: 31685637)

4. Address the concern that the PSD's territory may not be fully sampled in EM analyses. One suggestion would be for you to estimate the "Major axis" from the PSD in the EM reconstruction, which is how you summarized PSD size from single sections. This should help bridge the single section results with reconstructed psd data. Please also plot and report PSD surface area and ribbon volume for modiolar synapses that have only a single ribbon to be able to assess whether the increase in PSD surface area is caused by the higher frequency of synapses with multiple ribbons.

5. You should also indicate if the dendrite (sgn afferent) was sufficiently sampled. Do the boundaries of the PSD represent the last section in a series or were there additional sections that show the PSD was not present?

6. Please provide zoomed views of exemplar pre- and post-synaptic labeling on the Pillar and Modiolar sides, for wt and ko samples. This will better illustrate the primary IF data.

7. The possibility that GluA1 may compensate for the loss of GluA3 in inner hair cells needs to be discussed as this scenario cannot be excluded from the presented data given that GluA3KO IHCs exhibited a larger number of ribbons that lacked GluA2 and 4 (lone or 'orphaned' ribbons; Figure 6B).

8. Please clearly state potential exclusion criteria for the synapses that were sampled and analyzed by EM.

*Reviewer #1 (Recommendations for the authors):*

Points for the Intro and Discussion.

1) Intro. It needs to be stated early in the MS that older GluA3ko mice express behavioral deficits (changes in ABRs).

2) The description "pre-symptomatic, but in a pathological state.." is very useful, and I think the Intro and Discussion need to create more context for this idea. Though there may not be a plethora of background literature on the regulation of GluA expression at ihc-sgns to cite, the Intro does not give me a good sense of a) development, b) circuitry or c) pathology of "cochlear ribbon synapses".

3) Background literature downstream from the cochlea should be scaled down in the Intro and transferred to Discussion. This will create more room to develop a context for ihc ribbon-sgn afferent synapses; i.e. the differentiation of the pillar-modiolar axis.

4) In the Discission, there is not enough speculation on how the molecular changes and morphological changes (SVs) will shape the physiology. It would be interesting to know what you would predict to see when making whole-cell recordings from afferents, or if you were to use a GluSniffer.

5) For instance, you have previously shown the majority of spontaneous ihc-bouton epscs have a simple monophasic waveform in adult/3wk old mice (Becker et al 2018, eLife. Ribeye-ko). In your current study, you show the expression of GluA4 alone at certain ihc-sgn synapses of the GluA3ko mice. How might changes in epsc amplitude and kinetics be influenced by changes to receptor composition (a transition from GluA3/GluA2/A4 rich afferents to A4 alone in some instances)?

6) Can you speculate on whether "in general" changes in SV size (based on EM) may be more common on the pillar than the modiolar side when ihc synapses are perturbed? If you have some examples, beyond the excellent data you offer in the current study, please share. I note recent studies on endophilin kos (EMBO J, Kroll et al. 2019) and the AP180 ko (JCS, Kroll et al 2019) that have identified increased SV size as a ko phenotype, though a distinction in the pillar and modiolar was not made. Another set of examples comes from ribeye ko mice. Deletion of ribbons appears to have no influence on SV size (Becker 2018 PNAS), or it causes an increase in the diameter of membrane-proximal SVs (Philippe et al 2019 pnas); however, again neither study distinguished between the pillar and modiolar sides. To further confound our understanding of SV size regulation is that the conventional protocol for patching ihcs is to approach from the modiolar side; thus, changes in SV size on pillar side would be missed. For example, on-cell Cm measurements of SV size from Cav1.3 ko mice (from the modiolar side) showed no change in the size of SV size (membrane capacitance) when compared to wt ihcs (Grabner and Moser 2018, PNAS), albeit the synapses were severely affected, degenerated in the ko. It is possible the pillar side would have been more interesting to record from in the Cav ko, or alternatively, changes in SV size may be under more specific regulation and less associated with the severity of a perturbation on synapse maintenance. the

7) Lastly, you refer to nano-columns in the Discussion, lines 664/665, but in the case of ihc-sgn synapses, the presynaptic release machinery seems to be corralled into the center of a ring (fence) of postsynaptic GluA's. From the modeling study of Chapochnikov et al 2014 (Neuron), based in part on actual morphological data, the position of release is central to the ring of GluA receptors. This seems to be a clever design for optimizing SV readout. However, do you think that only the ribbon is central, while more primed SVs are docked lateral away the ribbon, directly above GkuAs? Is there any data that indicates the ihc AZ is ring shaped?

*Reviewer #2 (Recommendations for the authors):*

I would like to pass on a few comments/questions directly to the authors of the study.

I would be grateful if it was possible to get a more detailed account of some of the experimental procedures:

- Mouse breeding: Where were littermate C57BL6 (GluA3WT) and GluA3-knockout (GluA3KO) animals used or were the two colonies kept separately?

– Ultrastructural analyses: How were vesicle diameters measured? Was a correction performed to account for differences in section thickness for 3D reconstructions? Was the experimenter blinded for the genotype during image acquisition and analysis? How were synapses chosen for reconstruction? What was the pixel size?

With regards to statistical analyses, I would find it helpful if N- and n- numbers for the different groups and statistical tests used were listed in all Figure legends wherever relevant. It should also be clearly stated whether SD or SEM is shown in individual panels [especially if they appear to differ from what is mentioned in the text as it appears to be the case for the data shown in Figure 3 and 4 (ribbon volume)].

This may be a misunderstanding, but – as already mentioned in the public review- I would be grateful for clarification with regards to the data presented in Figure 3+4:

– p.16 l. 387: "A total of 27 synapses of GluA3WT mice were analyzed in 3D using serial sections (on average, 7 ultrathin sections per PSD). Of this total, 18 were on the modiolar side and 9 on the pillar side of the IHCs (Figure 3A-B). […] Six of 18 modiolar-side synapses had 2 ribbons (33.3%), whereas all the synapses on the pillar side had only 1 ribbon." In Figure 3C left, only 13 data points are shown for the PSD surface area and 17 for the ribbon volume for the modiolar side.

– p.17 l. 391: "Presynaptic ribbon volume was similar between these two groups (p = 0.525, Mann Whitney U test, U: 64; modiolar mean: 0.003 + 0.001 μm3; pillar mean: 0.002 + 0.001 μm3; Figure 3C, right"). In Figure 3 right, the box indicating the modiolar mean appears to be lower than 0.003.

– p.17 l. 406: "From GluA3KO, a total of 30 synapses were analyzed in 3D with serial sections (on average, 7 ultrathin sections per PSD). Of this total, 20 were on the modiolar side and 10 on the pillar side of the IHCs (Figure 4A-B). […] Six of the 20 modiolar-side synapses had 2 ribbons (30%) whereas all synapses on the pillar side had only 1 ribbon." Figure 4C left shows only 16 data points for the PSD surface area and 20 for the ribbon volume for the modiolar side.

– p.17 l. 408: "Analysis showed that PSD surface area and ribbon volume were similar for modiolar- and pillar-side synapses ([…] mean ribbon volume, modiolar: 0.003 + 0.001 μm3, pillar: 0.003 + 0.001 μm3; Figure 4C)." Despite the same reported mean, the height of the boxes in Figure 4C right is different (likely due to rounding). Would it be possible to present the data in a different way to take this into consideration?

Figure 3+4: What n-numbers were used for statistical testing and what do they refer to in the individual plots (number of PSDs analyzed, ribbons, vesicles)? Why did the authors choose to show the data for ribbon circularity as a Whisker plot and not the individual data points? It appears from the SV size (=diameter?) analysis, that the vesicles were pooled over all synapses analyzed for a given genotype. I would suggest comparing the mean vesicle diameter per synapse instead.

Would it be possible for the authors to also plot and report PSD surface area and ribbon volume for modiolar synapses that have only a single ribbon to be able to assess whether the increase in PSD surface area is caused by the higher frequency of synapses with multiple ribbons?

I would like to ask the authors to please show an overview of all reconstructed synapses as supplementary Figures (representative cross-section and rendering of the segmentation) similar to what has been shown in Payne et al., Front. In Synaptic Neuroscience, 2021. This would help to get a better idea about the variability in the size and morphology of the different synapses that went into the analysis.

Figure 5 A: "Data" instead of "Datata".

---

## [Author Response]

Essential revisions:1. We suggest a supplemental table summarizing the data and the statistical tests used for each of the analyses

We have now included with the revision the source data files for Figures 1, 3, 4, 5, 6, and 8.

2. There is more background signal in wild type than ko ihcs in Figure 2A "sum". Is there an explanation for this? Please also address the GluA3 staining in the GluA3 KO (Figure 2C).

Although analysis is always done on raw data, the displayed images were adjusted by changing the brightness and the contrast linearly. We have now added an explanation of this to the Methods on p. 27-28 of the revised text. Each wt/ko pair of image panels was adjusted to the same degree, but because each original image differed somewhat in non-specific background fluorescence of the tissue, the result is as you observed. In the initial submission, we failed to check the appearance of the images closely after conversion to pdf, which changes the appearance. Because these adjustments are made for clarity, for display purposes only, we have now taken the liberty in the revised document to re-adjust the brightness and contrast to avoid this appearance. We prefer to keep the re-adjusted version in the manuscript rather than the unadjusted version, for visual clarity.

Regarding the labelling in the KO on the GluA3 channel, there is indeed a small signal presumably due to cross-reactivity of the anti-GluA3 with GluA2 subunits, because the *C*-terminal epitope recognized by the GluA3 polyclonal antibody is in a region of high similarity between GluA2 and GluA3 (Dong et al., 1997). This relatively small cross-reactivity was approximately 10%, as quantified in Author response image 1.

**Author response image 1. sa2fig1:** 

The cross-reactivity in the KO is only apparent to the eye when the brightness level was over-adjusted to saturate the GluA3 signal in the WT, as in the original submission. Upon re-adjustment, the signal is not visually apparent. We also note, in the initial characterization of the GluA3 KO mice, García-Hernández et al., 2017 used southern blot and western blot to show a complete lack of Gria3 DNA or GluA3 subunit protein expression. Moreover, no labeling for GluA3 was found on auditory nerve synapses on bushy cells of GluA3KO mice using freeze-fracture immunogold labeling (Rubio et al., 2017). In both studies, we used an N-terminus antibody against GluA3. We now state this in the Methods on p. 28. Also, since approximately 10% of the signal quantified as GluA3 in the WT is putative cross-reactivity with GluA2, we now state this on p. 27-28.

3. Revise the text to emphasize your IHC data, rather than the brainstem. Reviewers would like to read a discussion of your findings in the context of other studies of morphological/functional changes of synapses upon deletion AMPAR subunits (e.g. PMID: 19409270; PMID: 23664612; PMID: 31685637)

We thank the reviewer for this suggestion. The AMPAR literature includes many experimental paradigms and settings that are potentially relevant to our study. However, we prefer to keep the text about GluA subunits in the auditory brainstem because we believe this provides the most appropriate context for the current study about GluA subunits in the auditory periphery. The 3 references mentioned by the reviewer are quite interesting stories involving AMPAR subunit composition, but we don’t think they are the most relevant papers to cite in the current study: PMID 19409270 and 23664612 are about the hippocampus; PMID 31685637 is about activity-dependent plasticity in the cerebellum. We would prefer to include such comparisons to other systems in future review articles. However, we did incorporate PMID 19409270 (Lu et al., 2009) and PMID: 19906877 (Lee et al., 2010) on p. 20 in Discussion, describing the predominance of GluA1/2 heteromers on CA1 cells in the hippocampus and the dominant influence of GluA1 on AMPAR plasticity in the brain. We contrast this with the mature cochlea, where GluA1 subunits are absent (Niedzielski and Wenthold, 1995; Matsubara et al., 1996; Parks, 2000; Shrestha et al., 2018) and the rules governing receptor dynamics are largely unknown and likely to be unique. As well, in response to the specific comments below, we revised the Intro and Discussion to enhance the context as suggested.

4. Address the concern that the PSD's territory may not be fully sampled in EM analyses. One suggestion would be for you to estimate the "Major axis" from the PSD in the EM reconstruction, which is how you summarized PSD size from single sections. This should help bridge the single section results with reconstructed psd data. Please also plot and report PSD surface area and ribbon volume for modiolar synapses that have only a single ribbon to be able to assess whether the increase in PSD surface area is caused by the higher frequency of synapses with multiple ribbons.

In response to the first concern, we have now supplemented the analysis with measurements of the linear length of the PSD (Figure 3, 4, and 5; see details in supplemental material Source data 1 for Figures 1,3,4, and 5). We have also revised all the 3D reconstructions and fixed some misalignments in some of the reconstructions. Overall, our data show a large variability of PSD sizes, especially in the synapses of the modiolar side (Figures 3, 4, and 5; see Figure 3—figure supplement 1 and Figure 4—figure supplement 1).

In response to the second concern, we have now assessed the data with synapses of double ribbons segregated from the data of synapses with single ribbons (Figures 3, 4 and 5); details in supplemental source data files: Figure 3-source data-1; Figure 4-source data-1; Figure 5-source data-1.

5. You should also indicate if the dendrite (sgn afferent) was sufficiently sampled. Do the boundaries of the PSD represent the last section in a series or were there additional sections that show the PSD was not present?

We added more details of the 3D reconstructions (p.32-33). In the serial ultrathin sections, we identified and photographed each synapse, but we only reconstructed the serial ultrathin sections that show a visible PSD; the images of the dendrites without a PSD were not included. The last section in a series represents the boundaries of the PSD.

6. Please provide zoomed views of exemplar pre- and post-synaptic labeling on the Pillar and Modiolar sides, for wt and ko samples. This will better illustrate the primary IF data.

Done, see new Figure 8—figure supplement 1.

7. The possibility that GluA1 may compensate for the loss of GluA3 in inner hair cells needs to be discussed as this scenario cannot be excluded from the presented data given that GluA3KO IHCs exhibited a larger number of ribbons that lacked GluA2 and 4 (lone or 'orphaned' ribbons; Figure 6B).

We thank the reviewer for this comment. We would like to explain our reasoning. As can be seen from the subpanels on the right side of Figure 6B, the occurrence of ribbons paired with GluA2 or GluA4 alone is about 10x less frequent than the occurrence of lone ribbons, making it unlikely that a GluA1 alone category would account for the lone ribbons, unless a mechanism exists to specifically express GluA1 alone in the absence of GluA3. However, we think the possibility of the apparently lone ribbons actually belonging to synapses expressing GluA1 alone is very unlikely given the data presented in Figure 1B showing lack of expression of GluA1 in the SGNs of GluA3 KO.

8. Please clearly state potential exclusion criteria for the synapses that were sampled and analyzed by EM.

In our ultrastructural analysis, we only included IHC-ribbon synapses that were clearly located on the modiolar or pillar sides of the IHC. We added this statement on p. 32-33.

Reviewer #1 (Recommendations for the authors):Points for the Intro and Discussion.1) Intro. It needs to be stated early in the MS that older GluA3ko mice express behavioral deficits (changes in ABRs).

We have now included this in the Introduction on p. 6.

2) The description "pre-symptomatic, but in a pathological state.." is very useful, and I think the Intro and Discussion need to create more context for this idea. Though there may not be a plethora of background literature on the regulation of GluA expression at ihc-sgns to cite, the Intro does not give me a good sense of a) development, b) circuitry or c) pathology of "cochlear ribbon synapses".

We agree. Please see revisions in the Introduction and Discussion, where we added more background about cochlear ribbon synapses with your comments in mind.

3) Background literature downstream from the cochlea should be scaled down in the Intro and transferred to Discussion. This will create more room to develop a context for ihc ribbon-sgn afferent synapses; i.e. the differentiation of the pillar-modiolar axis.

We appreciate the reviewer’s comments, but we believe that the literature of the auditory brainstem is necessary to understand the rationale of our study in the cochlea. Thus, we prefer not to scale it down, however, we have added more information about the ribbon synapse in the Introduction and Discussion. As result, the number of words in both sections has increased (Introduction from 693 to 851; Discussion from 1524 to 2,374).

4) In the Discussion, there is not enough speculation on how the molecular changes and morphological changes (SVs) will shape the physiology. It would be interesting to know what you would predict to see when making whole-cell recordings from afferents, or if you were to use a GluSniffer.

We agree. Please see the revisions in the Discussion on p. 21-22.

5) For instance, you have previously shown the majority of spontaneous ihc-bouton epscs have a simple monophasic waveform in adult/3wk old mice (Becker et al 2018, eLife. Ribeye-ko). In your current study, you show the expression of GluA4 alone at certain ihc-sgn synapses of the GluA3ko mice. How might changes in epsc amplitude and kinetics be influenced by changes to receptor composition (a transition from GluA3/GluA2/A4 rich afferents to A4 alone in some instances)?

We intend to perform patch clamp recordings in a follow-up study, and we now include some speculation of that in Discussion on p. 19-20. The GluA4 alone synapses may be transient, representing a terminal stage of pathology. Although they would be nearly impossible to sample with certainty in patch clamp given their very low frequency, we could speculate about changes due to the lack of GluA3 based on those changes observed in the endbulb synapses of the brainstem in Antunes et al., 2020. In that study, we found the presence of GluA3 resulted in fast AMPAR desensitization kinetics. As well, we found that GluA3 is required for the normal function, structure, and development of the presynaptic terminal, where the KO leads to altered short term-depression. The presence of GluA3 reduces and slows synaptic depression by lowering the probability of vesicle release, promoting efficient vesicle replenishment, and increasing the readily releasable pool of synaptic vesicles. GluA3 also makes the speed of synaptic depression rate-invariant. Therefore, we have added sentences to the Discussion on p. 21.

6) Can you speculate on whether "in general" changes in SV size (based on EM) may be more common on the pillar than the modiolar side when ihc synapses are perturbed? If you have some examples, beyond the excellent data you offer in the current study, please share. I note recent studies on endophilin kos (EMBO J, Kroll et al. 2019) and the AP180 ko (JCS, Kroll et al 2019) that have identified increased SV size as a ko phenotype, though a distinction in the pillar and modiolar was not made. Another set of examples comes from ribeye ko mice. Deletion of ribbons appears to have no influence on SV size (Becker 2018 PNAS), or it causes an increase in the diameter of membrane-proximal SVs (Philippe et al 2019 pnas); however, again neither study distinguished between the pillar and modiolar sides. To further confound our understanding of SV size regulation is that the conventional protocol for patching ihcs is to approach from the modiolar side; thus, changes in SV size on pillar side would be missed. For example, on-cell Cm measurements of SV size from Cav1.3 ko mice (from the modiolar side) showed no change in the size of SV size (membrane capacitance) when compared to wt ihcs (Grabner and Moser 2018, PNAS), albeit the synapses were severely affected, degenerated in the ko. It is possible the pillar side would have been more interesting to record from in the Cav ko, or alternatively, changes in SV size may be under more specific regulation and less associated with the severity of a perturbation on synapse maintenance.

These are interesting possibilities. We are unable to speculate as to whether or not changes in SV size may be more common on the pillar side. However, we did add the Kroll citations on p. 22 of the Discussion.

7) Lastly, you refer to nano-columns in the Discussion, lines 664/665, but in the case of ihc-sgn synapses, the presynaptic release machinery seems to be corralled into the center of a ring (fence) of postsynaptic GluA's. From the modeling study of Chapochnikov et al 2014 (Neuron), based in part on actual morphological data, the position of release is central to the ring of GluA receptors. This seems to be a clever design for optimizing SV readout. However, do you think that only the ribbon is central, while more primed SVs are docked lateral away the ribbon, directly above GkuAs? Is there any data that indicates the ihc AZ is ring shaped?

We are not aware of any data indicating a ring-shape of the presynaptic AZ. Based on the positions of morphologically docked SVs at the ribbon synapses in C57BL/6 WT mice, there are potentially primed SVs docked away from the ribbons, across from the ring of AMPARs (e.g., Payne et al., 2021 and many others). We now state on p. 23-24 that the existence of nanocolumns in the ribbon synapse is unclear.

Reviewer #2 (Recommendations for the authors):I would like to pass on a few comments/questions directly to the authors of the study.I would be grateful if it was possible to get a more detailed account of some of the experimental procedures:– Mouse breeding: Where were littermate C57BL6 (GluA3WT) and GluA3-knockout (GluA3KO) animals used or were the two colonies kept separately?

We have added to the Methods on p. 25.

– Ultrastructural analyses: How were vesicle diameters measured? Was a correction performed to account for differences in section thickness for 3D reconstructions? Was the experimenter blinded for the genotype during image acquisition and analysis? How were synapses chosen for reconstruction? What was the pixel size?

We have added more information and magnification settings to the Methods on p. 32-33.

With regards to statistical analyses, I would find it helpful if N- and n- numbers for the different groups and statistical tests used were listed in all Figure legends wherever relevant. It should also be clearly stated whether SD or SEM is shown in individual panels [especially if they appear to differ from what is mentioned in the text as it appears to be the case for the data shown in Figure 3 and 4 (ribbon volume)].

Please see the new spreadsheet in the revised manuscript, the number of samples has been added to the methods and the results.

This may be a misunderstanding, but – as already mentioned in the public review- I would be grateful for clarification with regards to the data presented in Figure 3+4:– p.16 l. 387: "A total of 27 synapses of GluA3WT mice were analyzed in 3D using serial sections (on average, 7 ultrathin sections per PSD). Of this total, 18 were on the modiolar side and 9 on the pillar side of the IHCs (Figure 3A-B). […] Six of 18 modiolar-side synapses had 2 ribbons (33.3%), whereas all the synapses on the pillar side had only 1 ribbon." In Figure 3C left, only 13 data points are shown for the PSD surface area and 17 for the ribbon volume for the modiolar side.

We updated and clarified the data on the Methods (p. 32-33) and the results p. 9-10.

– p.17 l. 391: "Presynaptic ribbon volume was similar between these two groups (p = 0.525, Mann Whitney U test, U: 64; modiolar mean: 0.003 + 0.001 μm3; pillar mean: 0.002 + 0.001 μm3; Figure 3C, right"). In Figure 3 right, the box indicating the modiolar mean appears to be lower than 0.003.

We updated and clarified the data on p. 9-10.

– p.17 l. 406: "From GluA3KO, a total of 30 synapses were analyzed in 3D with serial sections (on average, 7 ultrathin sections per PSD). Of this total, 20 were on the modiolar side and 10 on the pillar side of the IHCs (Figure 4A-B). […] Six of the 20 modiolar-side synapses had 2 ribbons (30%) whereas all synapses on the pillar side had only 1 ribbon." Figure 4C left shows only 16 data points for the PSD surface area and 20 for the ribbon volume for the modiolar side.

We updated and clarified the data on p. 10-12. We did not sample every synapse in a given IHC, so this M/P ratio does not mean necessarily there are 2x as many synapses on the M side compared with the P side. We have added this statement on p.32-33.

– p.17 l. 408: "Analysis showed that PSD surface area and ribbon volume were similar for modiolar- and pillar-side synapses ([…] mean ribbon volume, modiolar: 0.003 + 0.001 μm3, pillar: 0.003 + 0.001 μm3; Figure 4C)." Despite the same reported mean, the height of the boxes in Figure 4C right is different (likely due to rounding). Would it be possible to present the data in a different way to take this into consideration?

We updated and clarified the data on p. 10-12.

Figure 3+4: What n-numbers were used for statistical testing and what do they refer to in the individual plots (number of PSDs analyzed, ribbons, vesicles)? Why did the authors choose to show the data for ribbon circularity as a Whisker plot and not the individual data points? It appears from the SV size (=diameter?) analysis, that the vesicles were pooled over all synapses analyzed for a given genotype. I would suggest comparing the mean vesicle diameter per synapse instead.

We updated and clarified the data in the Methods and in Results sections. The histogram of circularity data in Figures 3 and 4 now shows the individual data points. We agree with the reviewer and now we show the mean vesicle diameter per synapse (Figures 3, 4, and 5).

Would it be possible for the authors to also plot and report PSD surface area and ribbon volume for modiolar synapses that have only a single ribbon to be able to assess whether the increase in PSD surface area is caused by the higher frequency of synapses with multiple ribbons?

We added the data to Figures 3, 4, and 5. The data of synapses (PSDs) with one and two ribbons on the modiolar side have been separated.

I would like to ask the authors to please show an overview of all reconstructed synapses as supplementary Figures (representative cross-section and rendering of the segmentation) similar to what has been shown in Payne et al., Front. In Synaptic Neuroscience, 2021. This would help to get a better idea about the variability in the size and morphology of the different synapses that went into the analysis.

See supplemental Figure 3—figure supplement 1 and Figure 4—figure supplement 1.